# Gender Parity in Spain: Attainments and Remaining Challenges

Isabel Árbol-Pérez [ID] and Francisco Entrena-Durán *[ID]

Faculty of Political Sciences and Sociology, University of Granada, 18071 Granada, Spain; isabelarbol92@ugr.es
* Correspondence: fentrena@ugr.es

**Abstract:** The purpose of this article is to study the progress made in Spain in terms of gender parity and the challenges still pending to be achieved in this regard. To attain this objective, first of all, the authors review the successive legal regulations aimed at reaching gender equality that have been enacted in Spain. Furthermore, the considerations and findings made are based on the use of quantitative and qualitative methodologies. On the one hand, from a quantitative viewpoint, different statistical data provided mainly by the Spanish Statistics National Institute are analyzed. From these data, the authors prepare a set of tables and figures that allow them to show that, despite the undoubted legislative advances attained, clear gender inequalities continue in Spain. On the other hand, the authors base their assertions both on their participant observation and on a reinterpretation and reanalysis of the results of two previous qualitative researches. One of the most remarkable outcomes of the use of this qualitative methodology is the persistence in Spain of diverse signs of macho mentality. This persistence not only manifests itself among many men, it is also shared by a large number of women.

**Keywords:** Spain; gender equality legislation; gender inequalities; activity rates; employment rates; gender pay gap; gender roles; macho mentality

## 1. Introduction

As is well known, the fight for equal rights between women and men is closely related to the emergence and development of the international suffrage movement that claimed the right of women to vote. Suffragism, which led to great strides in the social and legal recognition of the equality of women and men, appeared in the United States in the late 1840s and became widely established in the United Kingdom. Then, from 1865, the movement spread to a large part of the European countries, including Spain. However, in the last years of the 19th century, suffragism did not have the same impulse and impact in Spain that it had had and was having in the United States or the United Kingdom. The Association for the Teaching of Women (*Asociación para la Enseñanza de la Mujer*), created in 1870 by the pedagogue and intellectual Fernando de-Castro-y-Pajares, was a Spanish educational project, whose purported mission was to offer middle-class Spanish women the opportunity to have access to effective academic and scientific education, which they had hitherto lacked. This association, of which the writer Concepción Arenal was a close collaborator (Franco-Rubio 2004), sponsored several feminine schools that had a key role in the progress and social promotion of women in Spain, and specifically, in improving their education and job training.

In Spain, the first organizations headed by women to join the fight to advance women's freedoms and rights were not fully integrated into the suffrage movement in Europe and the United States. Such organizations include the Autonomous Society of Barcelona's Women (*Sociedad Autónoma de Mujeres de Barcelona*), founded by Ángeles López-de-Ayala, which gave way to the Progressive Society for Women (*Sociedad Progresiva Femenina*) in 1897, the same year in which Belén-de-Sárraga also created the General Women's Association of Valencia (*Asociación General Femenina de Valencia*) and the Federation of Resistance Societies

of Malaga (*Federación Malagueña de Sociedades de Resistencia*). Precisely, it was Ángeles López-de-Ayala who organized, on 10 July 1910 in Barcelona, the first demonstration led by women in Spain demanding for the first-time political rights for women (Arce-Pinedo 2008).

In 1904, the Socialist Feminine Group (*Grupo Femenino Socialista*) was also created in Bilbao. Later, that same group was formed in Madrid, where it remained active between 1906 and 1914, although since 2010 it has been called the Socialist Feminine Grouping (*Agrupación Femenina Socialista*). The two aforementioned organizations were pioneers in female collective action (Moral-Vargas 2005).

Later, in 1918, the National Association of Spanish Women (*Asociación Nacional de Mujeres Españolas*) was officially born. This association defended, among others, issues such as the reform of the Civil Code, educational promotion and the right of women to exercise liberal professions. In this context, in the 1920s, there were a series of demands, raised in the European setting, which led to the recognition of the right to vote for women. In the Spanish case, this contributed to creating the conditions that made it possible for women to be elected deputies in Congress, in such a way that, in the Second Republic (1931–1939), Clara Campoamor, Margarita Nelken and Victoria Kent became the first women deputies in Spain. Furthermore, it was precisely during the Second Republic, in the general elections of 19 November 1933, when, for the first time in history, Spanish women had the right to vote and went to the polls under equal conditions (Escudero 2013). Previously, in the elections to the Constituent Congress of the Second Republic in 1931, women enjoyed passive suffrage (the right to be voted and elected) but not active suffrage (the right to vote).

In these circumstances, the situation of women began to change significantly. Thus, privileges that had been recognized until that moment exclusively to men were eliminated. In addition, women's access to public office was regulated, the right of female election and vote was achieved (as we have indicated above), and rights were recognized for women in the family and in marriage. Moreover, civil marriage was recognized, along with the equal rights of women with regard to the children custody; the crime of adultery, which applied only to women, was suppressed; and divorce by mutual agreement was legally allowed. Moreover, the State was forced to regulate female work and protecting motherhood. In this way, those legal clauses that until then had allowed the dismissal of women from work when they married or had children were forbidden, compulsory maternity insurance was established, and wage equality for both sexes was approved.

In the field of education, mixed schools were allowed for the coeducation of both sexes, domestic and religious matters were abolished and night schools were created for workers. Female illiteracy was significantly reduced. In Catalonia, this extended even further: the dispensation of contraceptives was allowed; abortion was decriminalized and legalized on 25 December 1936, at the same time that the abolition of regulated prostitution was decreed; and it was forbidden to hire women in works considered hazardous or hard. In addition, in those areas of Spain that remained loyal to the Republic after the military coup in 18 July 1936, the practice of induced abortion was decriminalized in 1937 during the Spanish Civil War (1936–1939), when Federica Montseny was Minister of Health (from November 1936 to mid-May 1937) and during the government headed by the Socialist Francisco Largo Caballero.

However, all this step-by-step progress towards the equal rights of Spanish women (which was undoubtedly strengthened during the Second Republic) was abruptly truncated after the triumph in the Civil War of General Franco on 1 April 1939 and the beginning of the dictatorial forty-year-long regime known as Francoism, a regime that only ended when Franco died on 20 November 1975.

Francoism involved a clear recoil and involution as regards the rights of Spanish women. Specifically, in 1944 the Law of Labor Contract that formally derogated the Republican Law with the same name of 1931 was approved. This law, together with the promulgation of the 'Fuero de los Españoles' (translatable something like the Jurisdiction of the Spanish people), constitute the corpus of Franco Labor Legislation, in which women do not enjoy autonomy to hire or claim their correct retributions, things that are in charge

of their husbands and parents. Besides, the social and legal values imposed in Spain during the Franco Dictatorship demanded a code of morality that established strict sexual behavior standards for women (but not for men). In this background, the opportunities for professional careers were restricted for women, and their functions as wives and (most important) mothers were potentiated and 'honored'. Married women could not have personal or particular assets that were not included in their dowry, nor could they accept inheritance, appear in court by themselves or hire employees. Divorce, contraception and abortion were also prohibited. Nevertheless, prostitution was in fact tolerated (Clark 1990). In addition, during Francoism, marriages had to be canonical (that is, made according to Catholic laws and regulations) even if only one of the members of the couple was Catholic, which meant that all marriages in Spain had to be sanctioned by the Catholic Church. As the Catholic Church prohibited divorce, marriage could solely be dissolved by an arduous and costly annulment procedure that was available only after a long series of administrative measures and therefore was accessible only for those social sectors with greater purchasing power.

This situation changed radically after Franco's death and the arrival of democracy, in such a way that studies on gender equality have not stopped increasing in Spain during the last decades. The issues addressed by these studies are so numerous that it would exceed the limits of this article to attempt a systematic examination of them. Among such issues, we highlight here the growing concern about gender violence suffered by women (Castellano-Burguillo 2021; Moreno-Sánchez and Márquez-Vázquez 2016; Ruiz-Pérez and Pastor-Moreno 2021), sexism (Borrell et al. 2010; De-Francisco-Heredero 2019; Rodríguez-Burbano et al. 2021;), laws and measures aimed at advancing towards equality between women and men (Albert-Márquez and Soto-Arteaga 2018; Bodelón 2010; Lombardo and León 2014), the role of women in domestic work and care (Fernández-Cordón and Tobío-Soler 2005, 2019; Glass and Fujimoto 1994; Sánchez-Herrero-Arbide et al. 2009; Tobío-Soler 2002), the relationship between the transmission of employment patterns over generations of women and the spread of women's employment (Martín-Palomo and Tobío-Soler 2018), masculinities and gender identity (Salazar 2012; Tobío-Soler et al. 2021), etc. Recently, studies have proliferated about the most negative impacts of the COVID-19 crisis on women, given their situation of inequality (Fernández-Galiño and Lousada-Arochena 2021; Hupkau and Victoria 2020; Ruiz-Pérez and Pastor-Moreno 2021; Solanas 2020).

Our work aims to take a general approach to the situation of gender parity in Spain. Particularly, the purpose of our article is to take stock of the social advances achieved in terms of gender equality in Spain after Franco's death in 1975 and the subsequent arrival of democracy. Moreover, we will try to identify the challenges still pending in this regard. To achieve this goal, changes in the legislative framework are identified and analyzed. We will also show, as the equality sought or promulgated by laws, contrasts with the facts and figures revealing the situations of women inequality and discrimination who still persist de facto in our country. In this regard, we clarify here that we have chosen to evaluate progress in gender parity by using data on activity rates, employment rates and gender pay gap. We adopt this analytical strategy because it allows us a complete approach to the evident processes of increasing the incorporation of women into the labor market that have been happening in Spain in recent decades. Thus, this approach implies that we consider different variables affected by these processes that are decisive in determining the position of women.

## 2. Materials and Methods

The purpose of this article is to grasp the progress made in Spain in terms of gender parity and the challenges still pending to be achieved in this regard. To attain this purpose, first of all, we review the successive legal regulations aimed at reaching gender equality that have been enacted in Spain, after the arrival of democracy, during the last four decades. Furthermore, in our research, we mix quantitative and quantitative methodologies. On the one hand, from a quantitative point of view, we analyze a series of statistical data provided

mainly by the Spanish Statistics National Institute (*Instituto Nacional de Estadística*). From these data, we will draw up a series of figures and tables that will allow us to analyze if undoubted legislative advances achieved are followed by gender equality in Spain or not. On the other hand, from a qualitative viewpoint, and in order to deepen the understanding of the motivations and cultural habits that explain and/or justify (that is, legitimize) the persistence in Spain of inequalities between men and women, we have done participant observation. Participant observation entails that the researcher tries to immerse him or herself in some way in the social daily life, in the problems and in the expectations of the population he is studying. In this regard, we clarify that the participant observation mentioned here was carried out during two previous investigations in which the second author of the current article was involved (Del Río-Lozano et al. 2013; Entrena-Durán et al. 2021). In line with this, in these investigations the researchers were involved in several of the activities carried out by significant members of the population studied; at the same time, they held different meetings with that population and gave informal talks to it. Based on these observations and the qualitative interviews that were made in the course of the aforementioned researches, in the current article we have taken up and reinterpreted the results of these two researches related to gender equality in Spain. All of this has allowed us to see how different signs of macho mentality continue to persist in our country (Daros 2014; Mirandé 2018; Pérez-Martínez et al. 2021), which not only manifests itself among many men but is also shared by a large number of women. In this way, these women would have internalized in their mentality and perception of what their function is what Bourdieu has typified as 'male domination' (Bourdieu 2001).

### 3. Results

#### 3.1. Legal Changes towards Gender Equality

After Franco's death in 1975, a gradual process of rights recovery for the Spanish people began. This process is known as the Spanish transition to democracy. In this background, from the last decade of the 20th century to the present, different laws, aimed at achieving equal rights for women and men, have been approved. Two of these laws, which have been pioneering in the international arena, are the "Organic Law 1/2004, of December 28, of Comprehensive Protection Measures against Gender Violence" (*Ley Orgánica 1/2004, de 28 de Diciembre, de Medidas de Protección Integral contra la Violencia de Género*) and the "Law for the effective equality of women and men", published on 23 March 2007. As its title indicates, Law 1/2004, of 28 December, is an organic law. In other words, it is a law that derives directly from the Spanish Constitution and serves for its better application. Both this law and the second of them, apart from promoting equality between men and women, try to alleviate and regulate the effects of gender inequality and, above all, in the case of the first of them, to prevent violence by men against women because of their feminine condition.

In this way, after the recovery of democracy in Spain, the change in the legal-social status of women has been especially significant. An example of this is the sizeable and permanent increase in the number of women enrolled in the generality of university degrees, as well as the noteworthy and continuous growth of the presence of women in the labor market, which was experienced in Spain from the first years of the arrival of democracy. So, by the end of the 1970s, 22% of adult Spanish women—still somewhat less than in Italy and Ireland—had entered the labor market. But, by 1984, this figure had risen to 33%, a level not unlike that of Italy or the Netherlands. However, women still accounted for less than a third of the whole workforce, and in some important sectors, such as banking, the figure was close to one-tenth.

Despite this, things were gradually evolving towards a better situation for in Spain. Thus, already in the mid-1970s, from the very beginning of the transition to democracy, there was a progressive spreading of feminist social attitudes and movements in this country. The discourses of these attitudes and movements tried to identify the causes of female exclusion and sought to create the bases to end it (Fernández-Fraile 2008). All this

happened in a setting that had been developing already in the last years of the Franco dictatorship, characterized by the increasing incorporation of women into the labor market, the gradual decrease in the birth rate, a boom in foreign tourism, the emigration abroad of many Spaniards, and educational and cultural expansion. As a result of all this, at the end of the 1970s, attitudes favorable to the incorporation, on an equal footing, of women into the generality of the areas of social and working life had become common in Spanish public opinion, particularly among the younger population. These Spanish social attitudes were at the same level as their equivalent in other countries of our European environment. In these circumstances, the main barrier for women to access a job was no longer public opinion but factors such as a high unemployment rate and a lack of part-time jobs.

At the educational level, women quickly reached parity with men, at least statistically. Thus, in 1983, approximately 46% of university enrollment in Spain was female, which meant that our country had the thirty-first percentage in this regard in the world, with levels of female university students similar to those of the majority of the other European countries (Clark 1990).

Nevertheless, progress on gender equality has been slow in many ways. Thus, it took until 1987 for a ruling by the Supreme Court of Spain to consider that a rape victim did not have to prove that she had fought against her aggressor to defend herself, to verify the truth of her complaint. Until that date, when that important court case was resolved, it was generally accepted that a woman victim of rape, unlike victims of other crimes, had to show that she had put up a 'heroic resistance' to make it clear that she had not somehow tempted the rapist nor had otherwise encouraged him to attack her (Clark 1990).

In any case, the progress made since the end of the Franco dictatorship in terms of equality between the sexes is evident. To a large extent, this has been possible since the beginning of the new democratic era due to the fact that the Spanish Constitution of 1978 itself establishes equality as a chief value in its article 1, paragraph 1, even affirming in article 14 that all people are equal before the law, which is why any type of discrimination based on place of birth, race, sex, religion, opinion or personal or social situation is forbidden.

The constitutional legislative framework, together with the democratization of social and political life that this has made possible, has created a very appropriate setting for Spain to have undergone key advances in terms of equality between men and women over the course of more than forty years since Franco passed away.

Although the principle of gender equality is included in national and regional plans, programs and policies that are being implemented in the democratic context, the fact is that its implementation in practice has been much more difficult (European Institute for Gender Equality 2019). One of the events that slowed progress on gender equality was the 2008 economic crisis since, as a consequence of it, 'austerity' policies were implemented without a gender perspective. For instance, the budget cuts made during that crisis, in the interests of the proclaimed 'austerity' and cutback of public spending, affected care policies and resulted in the reabsorption of certain care tasks by families and, definitively, on the part of women.

The new government that came to power in June 2018 made it clear from the outset that gender equality policy was once again high on its agenda. In this way, the actions of this government have been characterized by a series of Spanish public policies aimed at promoting gender equality. Within the framework of these actions, plans and programs have been developed in our country that promote gender equality at the central, regional, and, to a certain extent, local levels. The main objectives of these plans have been to address gender equality in the workplace; empower women; and prevent and combat, at all levels, all forms of gender-based violence against women. Among these plans, we mention here two policies that are closely related to what we are dealing with in this paper. They are the "Royal Decree-Law 6/2019, of March 1, on urgent measures to guarantee equal treatment and opportunities between women and men in employment and occupation (Real Decreto 2019), and the "Royal Decree 902/2020, of October 13, on equal pay between women and men (Real Decreto 2020). Likewise, a Christmas campaign

launched in December 2021 by the Spanish Ministry of Consumer Affairs can also be included within these policies aimed at achieving gender equality. Thus, with this campaign, the Ministry wants to raise awareness about the risk of reproducing sexist roles and stereotypes in childhood through the advertising of games and toys (La Moncloa 2021b).

In addition to this, it is worth noting that, during the COVID-19 health crisis, the socialist government has launched campaigns and published different documents aimed at increasing the degree of social awareness about the situation of greater vulnerability of women. Thus, it has been pointed out that women are suffering doubly the consequences of the pandemic with an overload of health work, essential services and care (they continue to do most of the domestic work and care of dependents, paid and unpaid, also assuming a greater mental burden derived from these), greater job insecurity and poverty, and increased risk of suffering gender-based violence (La Moncloa 2021a).

In this context, the spirit and impulse of previous laws aimed at achieving equality for women, promulgated during the socialist government of Rodríguez-Zapatero (2004–2011), have been retaken. This government promoted some of the most profound legal advances on gender equality in the post-Franco democratic period. One of the measures adopted by the Rodríguez-Zapatero administration, in line with promoting gender equality, is Order PRE/525/2005, of 7 March, approved by the Council of Ministers on 4 March of the same year. Thus, in this Order, measures to be adopted to favor gender equality were indicated, and actions were promoted to reduce inequality in the areas of employment, the company, the compatibility of work and family life, research, solidarity, sport and gender violence. Specifically, among other things, the Order promoted the hiring and promotion of women at the work level and spoke of the changes in the regulations necessary to tackle harassment and gender-based violence at work.

Within this same Council of Ministers, Order APU/526/2005, of 7 March, was also approved, which established the Plan for Gender Equality in the General State Administration. This Plan, in addition to the aforementioned, established gender parity in some administrative positions and quotas for the hiring of women in others. Likewise, in keeping with its name, the Plan aimed to study the situation of women in the General State Administration in relation to gender equality, with the aim of intervening to alleviate the inequalities that could be found.

However, the most important step towards gender equality was taken in Spain with the approval of Organic Law 3/2007 of 22 March, aimed at achieving effective equality between women and men (known as the Equality Law). This Law, which is applied at the national, regional and local levels, covers a wide range of issues, from paternity leave to a more balanced political gender representation, and establishes the duty, on the part of public bodies and companies with more than 250 employees, to develop equality plans in cooperation with workers' representatives. The Equality Law also prescribed the creation of gender units in all ministries.

The promulgation of Organic Law 3/2007 entails the recognition that, despite full constitutional recognition of equality before the law, it has not been achieved in matters of gender. Thus, it is possible to observe a clear wage discrimination between men and women, as well as in pensions charged by both sexes and in pensions for widows. Furthermore, unemployment levels are higher among the female population, while the presence of women in positions of responsibility at all levels (political, economic, social and cultural) is lower. In addition, women are discriminated against in terms of reconciling work and family life, since most of them are the ones who assume responsibility for taking care of the home, while a significant number of men tend to ignore it. Taking this situation into account, a priority objective of the Law was to carry out normative action to combat discrimination against women based on their sex, which continued to prevail either directly or indirectly. To achieve this purpose, it was necessary to remove those obstacles and stereotypes that prevented attaining parity between men and women. In particular, the Law contemplated special consideration for the most vulnerable groups of women, namely, minorities, migrants and women with some kind of disability.

The major novelties of this Law are that it tries to prevent discriminatory behavior against women and that it plans to carry out active policies to achieve equality. Specifically, the Law envisaged intervening at all levels (state, regional and local) through public policies and the setting up of criteria for action by all public powers. Such actions would be established on educational, health, artistic and cultural policies, as well as in the areas of the information society, rural development, housing, sports, culture, and spatial planning or development cooperation, among others.

In order to encourage the achievement of all this, the Law intended to apply a Strategic Plan for Equal Opportunities by creating an Inter-Ministerial Equality Commission. This Commission is responsible for coordinating, monitoring and preparing gender impact reports, as well as making periodic evaluations on the effectiveness of the actions carried out and meeting at least twice a year. The rhythm of actions and activities of this Commission have been irregular since it was created. However, with the coming to power of the current government of Spain, after a motion of censure, a new impetus was given to equality policies. An example of this is that on 7 November 2018, the Inter-ministerial Commission for Equality between women and men met, which had not met since 2011 (Reunión de la Comisión Interministerial 2018). Among other things, the Commission agreed to prepare the Royal Decree for the regulatory development of Equality Units and to improve the coordination of existing Units in all ministries. It also agreed to begin work aimed at the reformulation of a new Strategic Plan for Equal Opportunities between men and women. This Plan includes four fundamental pillars: incorporation of the gender perspective in a transversal way, new social pact, citizenship and violence against women (Tablado 2021).

Regarding the compatibility of people's work and family activity, Organic Law 3/2007 introduced an important novelty with respect to the law in force until then (Law 39/1999, of 5 December). Thus, a 13-day leave was now established for fathers that was not contemplated in the previous legislation, as well as the extension of two weeks of leave in the event that a child is born with a disability. These two weeks can be enjoyed indistinctly by a father or mother. Qualitatively, this entails crafting an adequate legislative framework to foster the greater participation of men in raising children than had previously been contemplated.

### 3.2. Facts and Figures on Gender Inequality in Spain

Having seen the progress on the legislative level, we consider that now is the time to analyze the existing data in this regard. All this with the purpose of assessing to what extent the formal equality between men and women considered in the aforementioned legislation has actually been reflected in everyday reality.

In the first place, there are two groups of data that we are going to take into consideration in this section, given their special usefulness to reflect how the gender equality that is being talked about here materializes in reality. On the one hand, we consider data on the activity rate and on the other the data on employment rates. We do all this in order to see the proportions of women who remain exclusively devoted to the traditional role assigned to them as caretakers of their homes and families, and to determine the proportion of those women who have entered the labor market.

Regarding the activity rate, it measures, in percentage, the ratio between the active population (employed and unemployed) and the overall population over 16 years of age (all those of working age, even if they do not work or seek job). In this regard, as can be seen in Figure 1, in Spain the activity rate of women remains below the activity rate of men. As can be seen in Figure 2, the considerably higher inactivity rate in the female population is due to the fact that many women are dedicated to the care of adults with disabilities, to children and other family, or to personal motives. This fact is closely related to the fact that there are still a large number of women who are neither working nor looking for work outside the home since they are devoted to the social role that has traditionally been assigned to the female gender as housewives and/or mothers (Glass and Fujimoto 1994; Nandini et al. 2019).

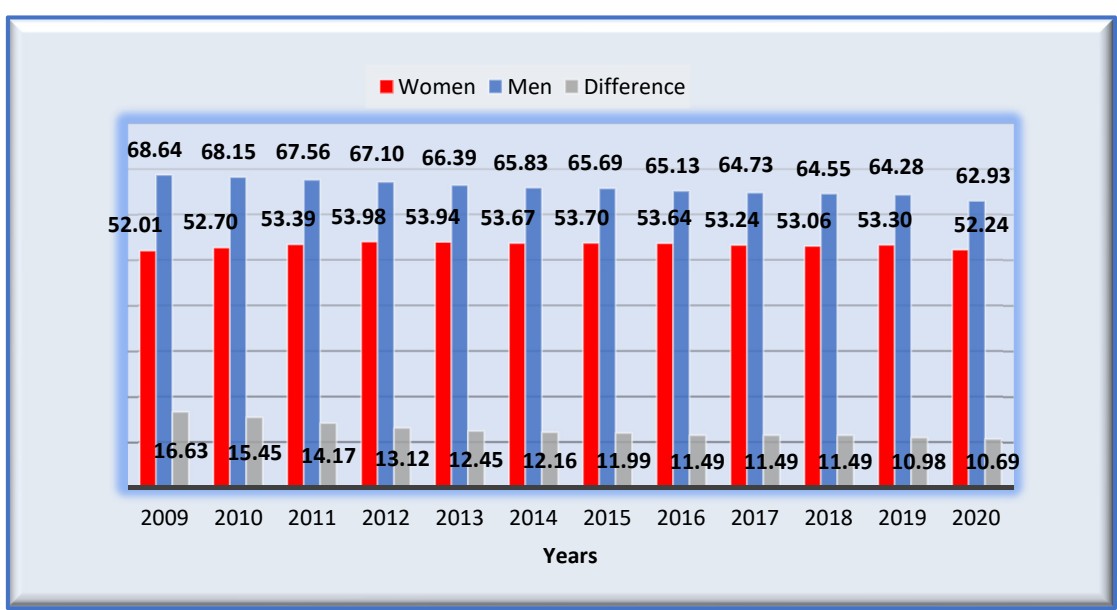

**Figure 1.** Activity rates by sex (percentages). Source: Authors with data from the Statistics National Institute (*Instituto Nacional de Estadística*).

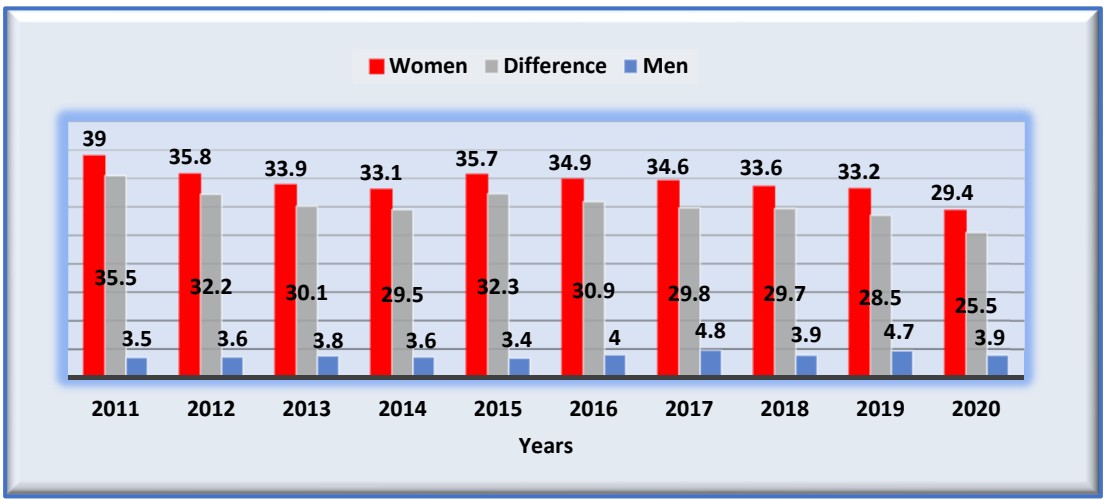

**Figure 2.** Inactive population in Spain from 15 to 64 years not seeking employment. Reason: care of adults with disabilities or children and other family or personal motives (percentages). Source: Authors with data from EUROSTAT (available at: http://appsso.eurostat.ec.europa.eu/nui/show. do?dataset=lfsa_igar&lang=en (accessed on 23 September 2021).

In any case, the evolutionary trend in recent years has been towards a gradual decrease in the gender gap in terms of activity rates. Thus, in the time period referred to in the data on which Figure 1 is based (2009–2020), the difference between the activity rates of men and the activity rates of women has decreased considerably, going from a difference of 16.63 points in 2009 to a difference of 10.69 points in 2020.

Regarding those people whose marital status is married (we do not have data on common-law couples who have not legally formalized their cohabitation situation), they show a trend similar to that mentioned above, as can be seen in Figure 3. However, it is possible to observe a more pronounced decrease in the difference between the activity rates of men and those of married women, in such a way that, while in 2009 this difference was 16.37 points, by 2020 it had been reduced to 8.46 points. What is remarkable is that this happened despite the fact that many of the married women were combining their unpaid domestic work with their job outside the home (Aguilar-Barceló and López-Pérez 2016;

Conejo-Pérez et al. 2021; Fernández-Cordón and Tobío-Soler 2005). Therefore, we have here a proof of the great effort that these women made to integrate into the labor market. Anyway, despite the evident advances in gender equality that this data evolution shows, we should not forget that there is currently a noticeable gap in the activity rates of women compared to those of men. This is an indicator that there are still many women on the fringes of the labor market, who are often not looking for work and tend to be exclusively committed to caring for their family and homes.

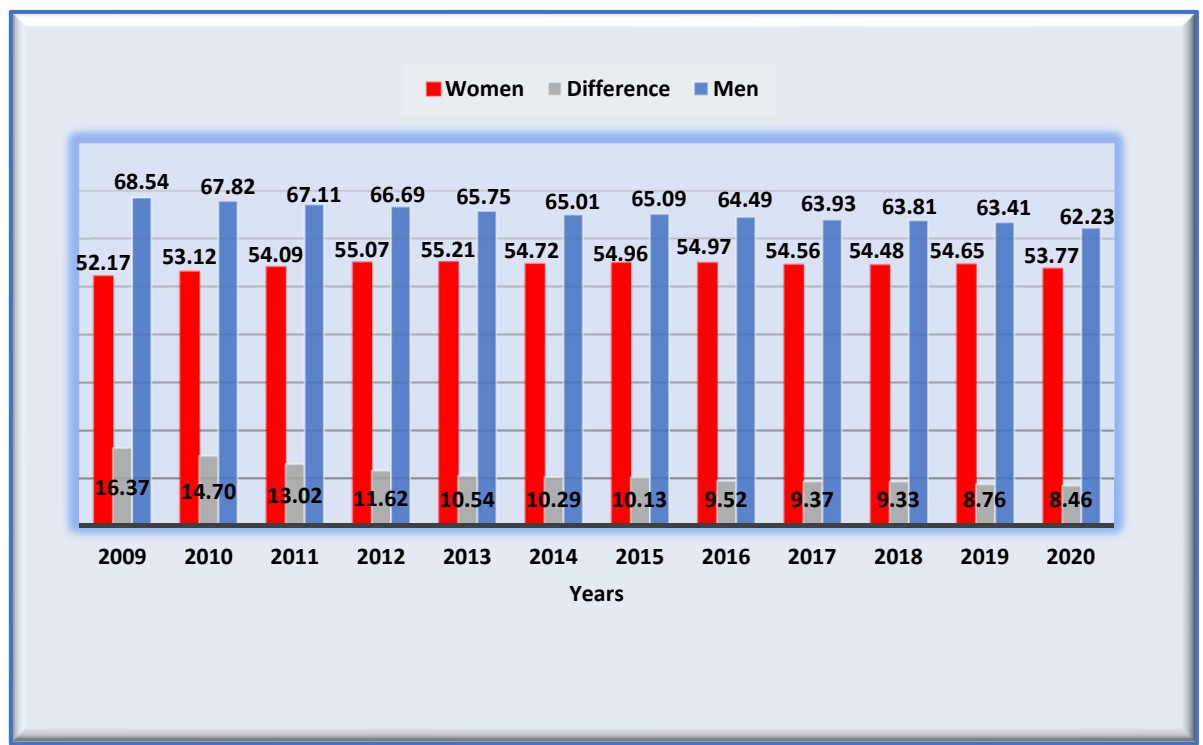

**Figure 3.** Activity rates of married people by sex (percentages). Source: Authors with data from the Statistics National Institute (*Instituto Nacional de Estadística*).

In contrast to the above, although as shown in Figure 4 differences between male and female activity rates are still present, such differences are drastically reduced in the case of separated or divorced persons. Thus, in the period considered in this figure, it can be observed that the activity rates of divorced or separated men and women almost overlap. Furthermore, except for the years 2010, 2011 and 2020, the differences between male and female activity rates are negative; that is, in most years of the period studied, the percentage of divorced or separated women who work or look for work outside the home is higher.

Could this higher proportion of separated or divorced women who work or look for work be understood in the sense that these women are more pushed to look for an employment because they do not have the financial support of a partner? Or, could this situation reveal that there are a considerable number of women who, precisely because they choose to enter the labor market, renounce (or, rather, they are compelled to renounce due to circumstances) marriage? Or are only financially independent women ready to get out of an unhappy marriage while financially dependent women choose to stay in an unhappy marriage because they do not have an alternative? Regardless of the diverse and particular concrete answers that, to explain the case of each specific woman, may be given to the aforementioned questions, the truth is that, in general, the affirmative answers to such questions are in line with the reality of the situation of women in Spain, which, we reiterate, continues to show the existence of clear gender gaps despite the undoubted legislative advances produced in recent decades.

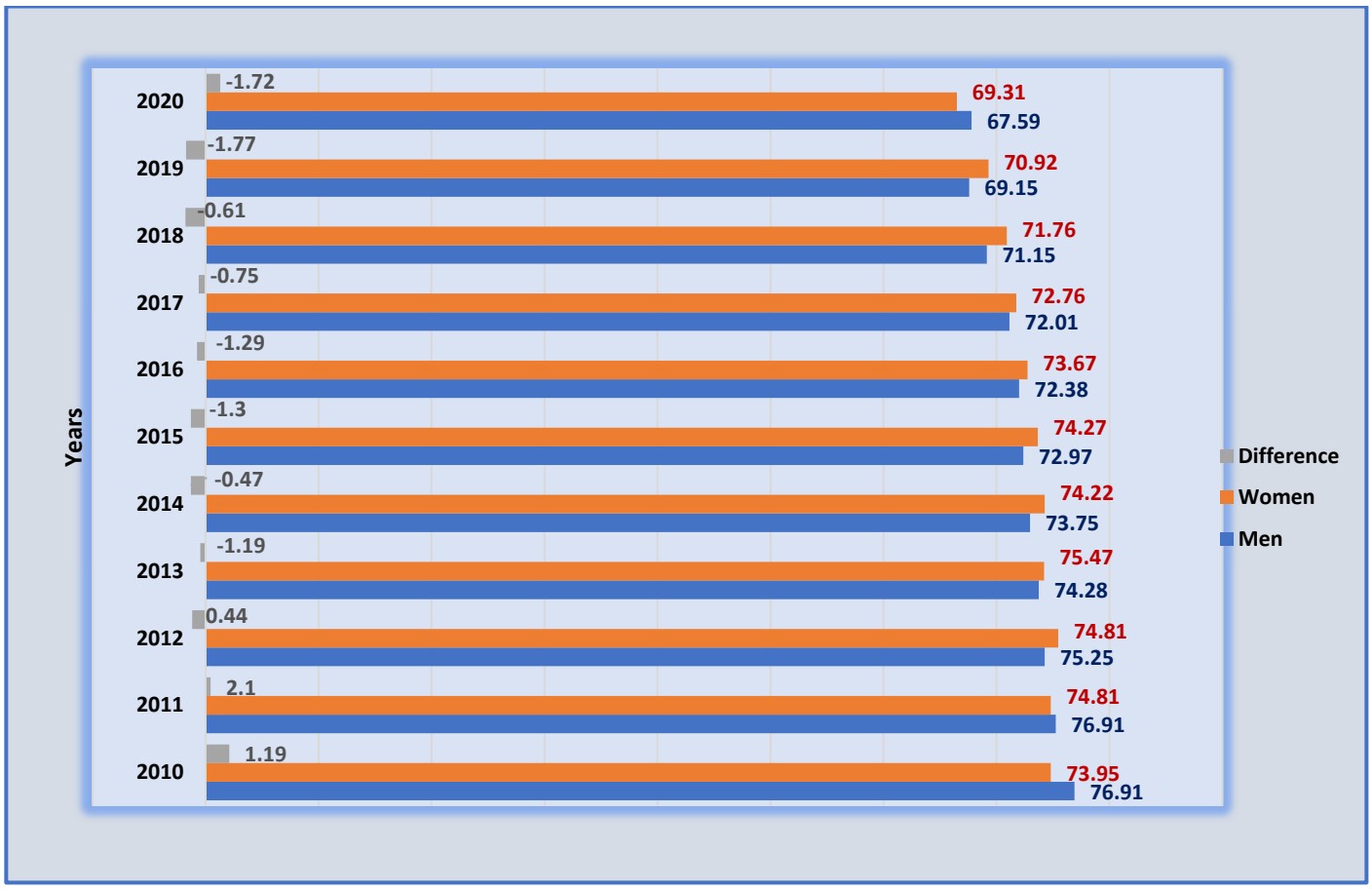

**Figure 4.** Activity rates of separated or divorced people by sex (percentages). Source: Authors with data from the Statistics National Institute (*Instituto Nacional de Estadística*.

Apart from what has been argued before, we take into account the employment rate, which is the ratio between, on the one hand, the number of people employed included in the age range from 16 to 64 years, and, on the other, the whole population of the same age range, that is, the working-age population. In this regard, it should be noted that, as can be seen in Figure 5, some progress is being made. Thus, despite the percentage of women employed being lower than those of men, it can be observed that the differences between the male and female employment percentages have been gradually reducing in the period referred to in this figure (2009–2020), in such a way that a gradual increase in the percentage of employed women has been experienced in this period, while at the same time there has been a trend towards a certain decrease in the percentage of employed men.

Regarding how male and female jobs are distributed by activity branches, Figure 6 shows that there are activity branches in which male work predominates, activity branches in which female jobs predominate, and activity branches in which there is a certain gender parity. By activity branches with a predominance of male population, we understand those in which men working represent more than 60% of employment in that branch. Correlatively, the activity branches in which female population prevails are those in which more than 60% of their jobs are held by women. Finally, the activity branches in which it can be considered that there is gender parity are those that show a clear balance in the distribution of jobs between men and women.

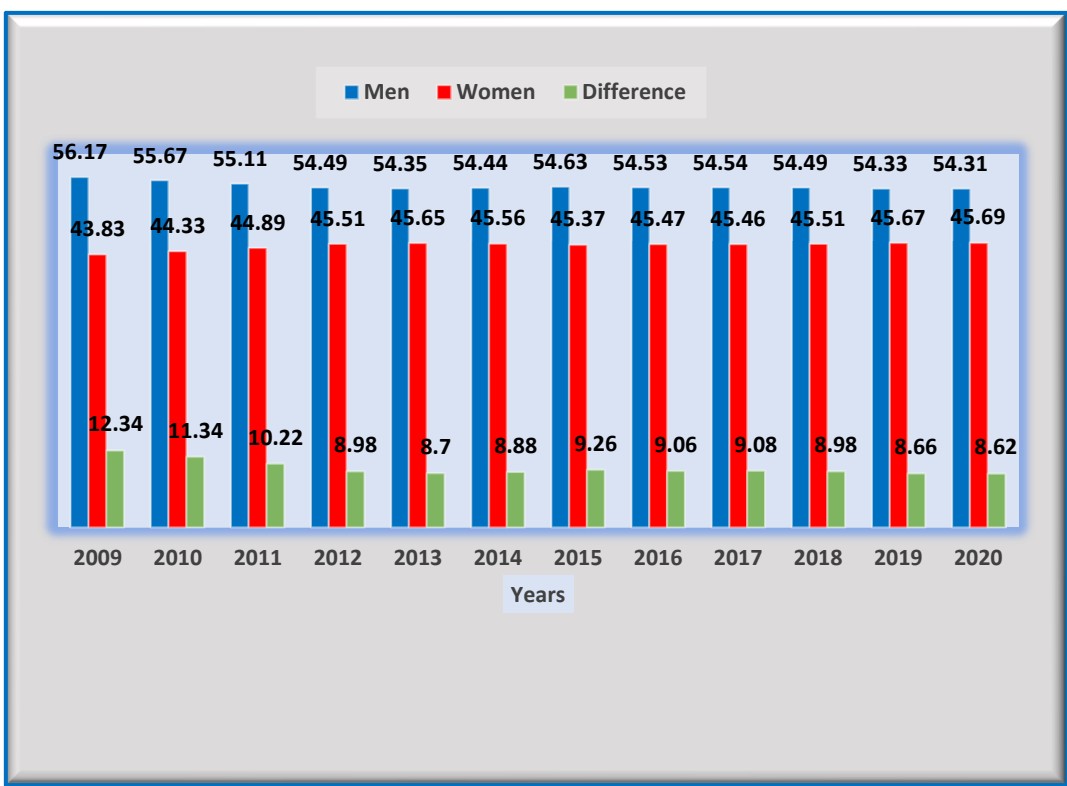

**Figure 5.** Employed persons by sex (percentages). Source: Authors with data from the Statistics National Institute (*Instituto Nacional de Estadística*).

Continuing with Figure 6, we must explain to the reader that each of the activity branches included in it is divided into different categories of data, which we have omitted. The reason for this exclusion has been to prevent this figure from being larger and more detailed than it already is since that would have made it difficult to understand it. We make this clarification given that next, in addition to commenting on Figure 6, we present a series of additional data and calculations that are not reflected in this figure; at the same time, we make several tables on that data.

First, Figure 6 shows that the activity branches in which the majority of jobs are male are building, extractive industries, transport and storage, agriculture, cattle raising, forestry and fishing, manufacturing industry, and information and communications. In this regard, based on the information in this figure, we have calculated the distribution of jobs between men and women within each of these activity branches in which male jobs predominate over female ones. We detail the percentages of that distribution in Table 1.

**Table 1.** Percentage distribution of employment by sectors in those activity branches with the highest proportion of male population.

| Activity Branch | Men | Women |
|---|---|---|
| Building | 91.13 | 8.87 |
| Extractive industries | 75 | 25 |
| Transport and storage | 76 | 24 |
| Agriculture, cattle raising, forestry and fishing | 74.36 | 25.64 |
| Manufacturing industry | 68.03 | 31.97 |
| Information and communications | 65 | 35 |

Source: Authors with data corresponding to the fourth quarter of 2020 provided by the Statistics National Institute (*Instituto Nacional de Estadística*).

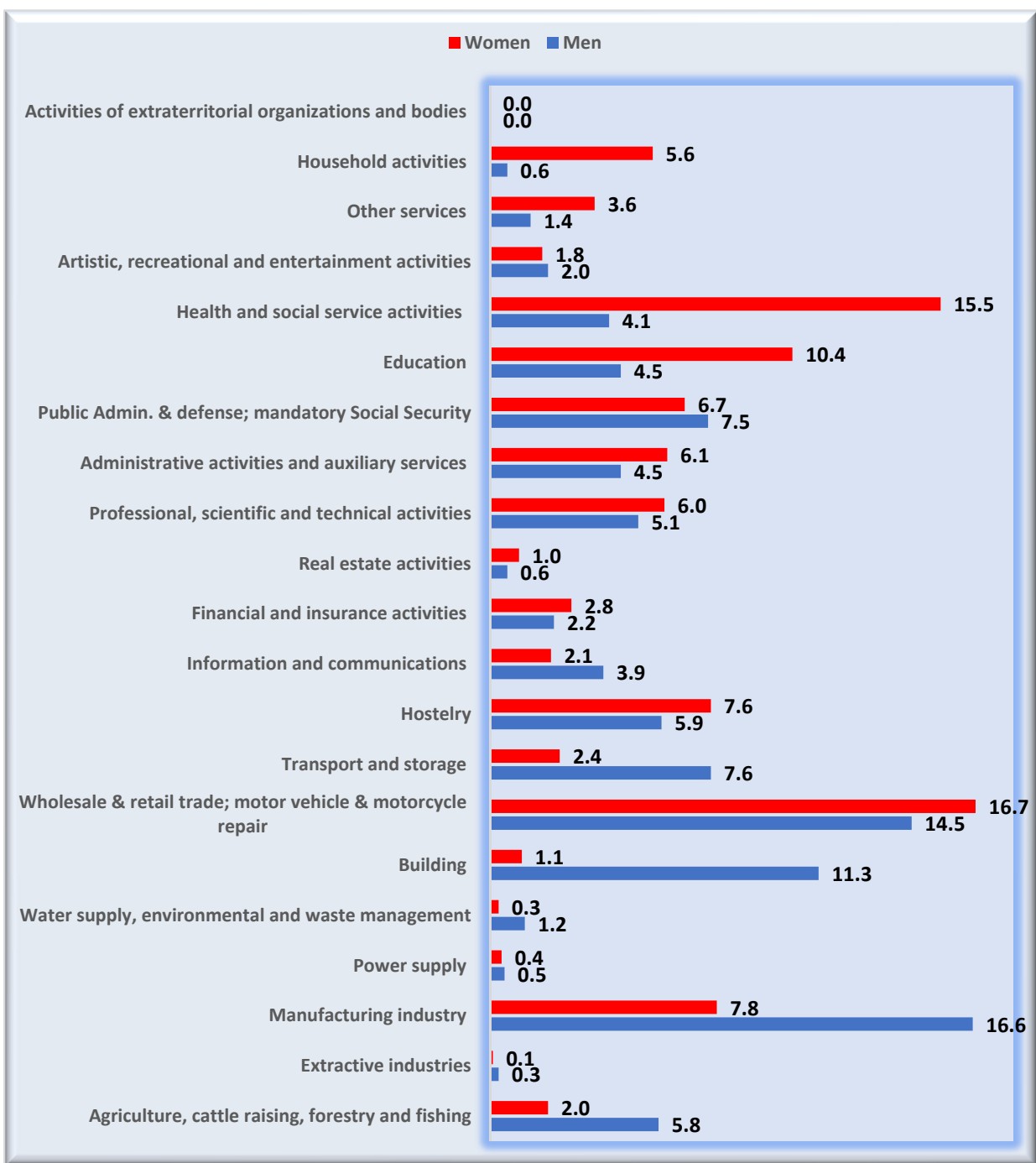

**Figure 6.** Working persons in Spain by sex and activity branch (percentages). Source: Authors with data corresponding to the fourth quarter of 2020 provided by the Statistics National Institute (*Instituto Nacional de Estadística*).

Secondly, also in Figure 6, it is observed that the most feminized activity branches are household activities, real estate activities, health and social service activities, education, and other services. In this regard, based on the information in this figure, we have calculated the distribution of jobs between men and women within each of these activity branches in which female jobs preponderate over male ones. We detail the percentages of that distribution in Table 2.

**Table 2.** Percentage distribution of employment by sectors in those activity branches with the highest proportion of female population.

| Activity Branch | Men | Women |
|---|---|---|
| Household activities | 9.68 | 90.32 |
| Real estate activities | 37.50 | 62.50 |
| Health and social service activities | 20.92 | 79.08 |
| Education | 30.20 | 69.80 |
| Other services | 28.00 | 72.00 |

Source: Authors with data corresponding to the fourth quarter of 2020 provided by the Statistics National Institute (*Instituto Nacional de Estadística*).

Third, the observation of Figure 6 shows us that there are a series of activity branches in which there is a certain gender parity in the distribution of employment. These are the following ones: professional, scientific and technical activities; wholesale and retail trade, motor vehicle and motorcycle repair; hostelry; financial and insurance activities; administrative activities and auxiliary services; public administration and defense; mandatory social security; power supply (supply of electricity, gas, steam and air conditioning); and artistic, recreational and entertainment activities. In this regard, based on the information in this figure, we have calculated the percentage distribution of jobs between men and women within each of these activity branches. The percentage calculations carried out, which we present in Table 3, show us that in these activity branches neither female nor male jobs exceed 60% in any case.

**Table 3.** Activity branches that show gender parity in the percentage distribution of employment.

| Activity Branch | Men | Women |
|---|---|---|
| Professional, scientific and technical activities | 45.95 | 54.05 |
| Wholesale and retail trade; motor vehicle and motorcycle repair | 46.47 | 53.53 |
| Hostelry | 43.70 | 56.30 |
| Financial and insurance activities | 44 | 56 |
| Administrative activities and auxiliary services | 42.45 | 57.55 |
| Public Administration and defense; mandatory Social Security | 52.82 | 47.18 |
| Power supply (supply of electricity, gas, steam and air conditioning) | 55.56 | 44.44 |
| Artistic, recreational and entertainment activities | 52.63 | 47.37 |

Source: Authors with data corresponding to the fourth quarter of 2020 provided by the Statistics National Institute (*Instituto Nacional de Estadística*).

As we have mentioned previously, the professions included in the activity branches considered in Figure 6 are very heterogeneous. Therefore, in addition to the aforementioned Tables 1–3, we now make a more detailed analysis of several of these branches, as well as in those cases in which the branch that we analyze in detail includes a number of categories of 3 or higher, where we create additional tables. All this is done in order to show how egalitarian the distribution of jobs really is within the different categories of activity that such activity branches include.

An activity branch that encompasses different categories within it is "Professional, scientific and technical activities". Thus, in this branch, as can be seen in Table 4, the following categories are contained: 1. legal and accounting activities; 2. central headquarters activities, business management consulting activities; 3. architectural and engineering technical services, technical testing and analysis; 4. research and development (R&D); 5. advertising and market research; 6. other professional, scientific and technical activities; and 7. veterinary activities. Among the aforementioned activities, only in that of "Architectural and engineering technical services; technical testing and analysis", male jobs are the majority (65.22% of jobs), while in the rest of the activities female employment is the majority, especially highlighting the case of veterinary activities, in which women occupy 66.67% of jobs compared to men who hold 33.33% of them.

**Table 4.** Percentage distribution of employment by sectors in the activity branch Professional, scientific and technical activities.

| Professional, Scientific and Technical Activities | Men | Women |
|---|---|---|
| Legal and accounting activities | 38.46 | 61.54 |
| Central headquarters activities; business management consulting activities | 42.86 | 57.14 |
| Architectural and engineering technical services; technical testing and analysis | 65.22 | 34.78 |
| Research and development (R&D) | 42.86 | 57.14 |
| Advertising and market research | 45.45 | 54.55 |
| Other professional, scientific and technical activities | 46.15 | 53.85 |
| Veterinary activities | 33.33 | 66.67 |

Source: Authors with data corresponding to the fourth quarter of 2020 provided by the Statistics National Institute (*Instituto Nacional de Estadística*).

Another activity branch worth commenting on in more detail is "Wholesale and retail trade; repair of vehicles and motorcycles". This branch includes the following categories of activities: 1. sale and repair of motor vehicles and motorcycles; 2. wholesale trade and trade intermediaries, except of motor vehicles and motorcycles; and 3. retail trade, except of motor vehicles and motorcycles. As can be seen in Table 5, both in category 1 and in category 2 employment is predominantly male. On the other hand, in category 3, female employment is the majority.

**Table 5.** Percentage distribution of employment by sectors in the activity branches wholesale and retail trade, and motor vehicle and motorcycle repair.

| Wholesale and Retail Trade; Motor Vehicle and Motorcycle Repair | Men | Women |
|---|---|---|
| Sale and repair of motor vehicles and motorcycles | 82.35 | 17.65 |
| Wholesale trade and trade intermediaries, except of motor vehicles and motorcycles | 60.26 | 39.74 |
| Retail trade, except of motor vehicles and motorcycles | 35 | 65 |

Source: Authors with data corresponding to the fourth quarter of 2020 provided by the Statistics National Institute (*Instituto Nacional de Estadística*).

As for the hostelry branch, as we have pointed out before, it is a feminized activity branch when it comes to the distribution of employment. Thus, the following activities are integrated into this branch: 1. accommodation services (40.74% of male employment compared to 59.26% of female employment); and 2. food and beverage services (44.44% of male employment compared to 55.56% female employment).

Regarding the branch "Financial and insurance activities", in Table 6 we see that it is composed of the following activities: 1. financial services, except insurance and pension funds; 2. insurance, reinsurance and pension funds, except mandatory Social Security; and 3. auxiliary activities to financial services and insurance. In these three activities, female jobs are the majority but in none of them does the percentage of jobs held by women reach 60%, while male jobs exceed 41% in all three cases. In short, it could be stated that within this activity branch the distribution of jobs between men and women is relatively balanced.

**Table 6.** Percentage distribution of employment by sectors in the activity branch Financial and insurance activities.

| Financial and Insurance Activities | Men | Women |
|---|---|---|
| Financial services, except insurance and pension funds | 44.44 | 55.56 |
| Insurance, reinsurance and pension funds, except mandatory Social Security | 41.18 | 58.82 |
| Auxiliary activities to financial services and insurance | 42.86 | 57.14 |

Source: Authors with data corresponding to the fourth quarter of 2020 provided by the Statistics National Institute (*Instituto Nacional de Estadística*).

With regard to "Administrative activities and auxiliary services", these are made up of the activities listed in Table 7. In this table, it can be seen that female employment

is predominant in most categories, namely, in "employment-related activities", "travel agencies activities", "services to buildings and gardening activities" and "administrative office activities and other auxiliary activities to companies". In contrast, male employment is only predominant in "rental activities" and in "security and investigation activities".

**Table 7.** Percentage distribution of employment by sectors in the activity branch "Administrative activities and auxiliary services".

| Administrative Activities and Auxiliary Services | Men | Women |
|---|---|---|
| Rental activities | 60.00 | 40.00 |
| Employment-related activities | 33.33 | 66.67 |
| Travel agencies activities | 33.33 | 66.67 |
| Security and investigation activities | 75.00 | 25.00 |
| Services to buildings and gardening activities | 35.94 | 64.06 |
| Administrative office activities and other auxiliary activities to companies | 33.33 | 66.67 |

Source: Authors with data corresponding to the fourth quarter of 2020 provided by the Statistics National Institute (*Instituto Nacional de Estadística*).

Finally, with reference to "Real Estate Activities" and "Public Administration and defense; mandatory Social Security", these activity branches do not include other activities within them. Therefore, we do not make any more comments about them here than we have previously made when we have discussed the data that is reflected in Figure 6.

As is well known, there is usually a relationship between the educational level attained and the distribution of people employed by sex. In this respect, Figure 7 shows that, except at superior education levels, where the number of employed women is markedly higher than that of men with the same educational level, as a general rule the number of employed men is greater than that of women employed at all educational levels. The smallest differences between men and women are found in upper secondary education levels (second stage of this education, either with general or professional orientation), while the greatest differences, as shown in the figure, are found at the level of the first stage of secondary education.

As can be seen in Figures 8 and 9, the relationship between the educational level attained and the distribution of employed persons by sex does not vary significantly when we distinguish between married employed persons and unmarried employed persons. In this way, similar to what Figure 7 shows, in Figures 8 and 9 it is observed that the number of men employed in all educational levels is greater, except in the case of higher education, in which there are more women employees than men. Furthermore, with respect to the unmarried employees (Figure 9), it is observed that the differences between employed men and women with the same educational levels are less than in the case of married employees, except for the cases of the second stage of secondary education (professional orientation) and especially in higher education, in which the number of unmarried women working is considerably higher. In sum, the analysis of the information provided by Figures 8 and 9 seems to contribute to validating the thesis according to which, as people's educational levels rise, gender parity increases in terms of job opportunities; moreover, such parity would tend to be higher among unmarried persons.

As can be seen in Table 8, in the cases of three economic sectors of great weight in employment and GDP such as industry, building and services, the average annual earnings per worker for women was lower than the average annual earnings for men in 2019. So, in these three cases there was an evident pay gap for women compared to men. The notable decrease in this gap in the building sector could be understood in the sense that in this sector there are very few women who work and most of them have high educational levels.

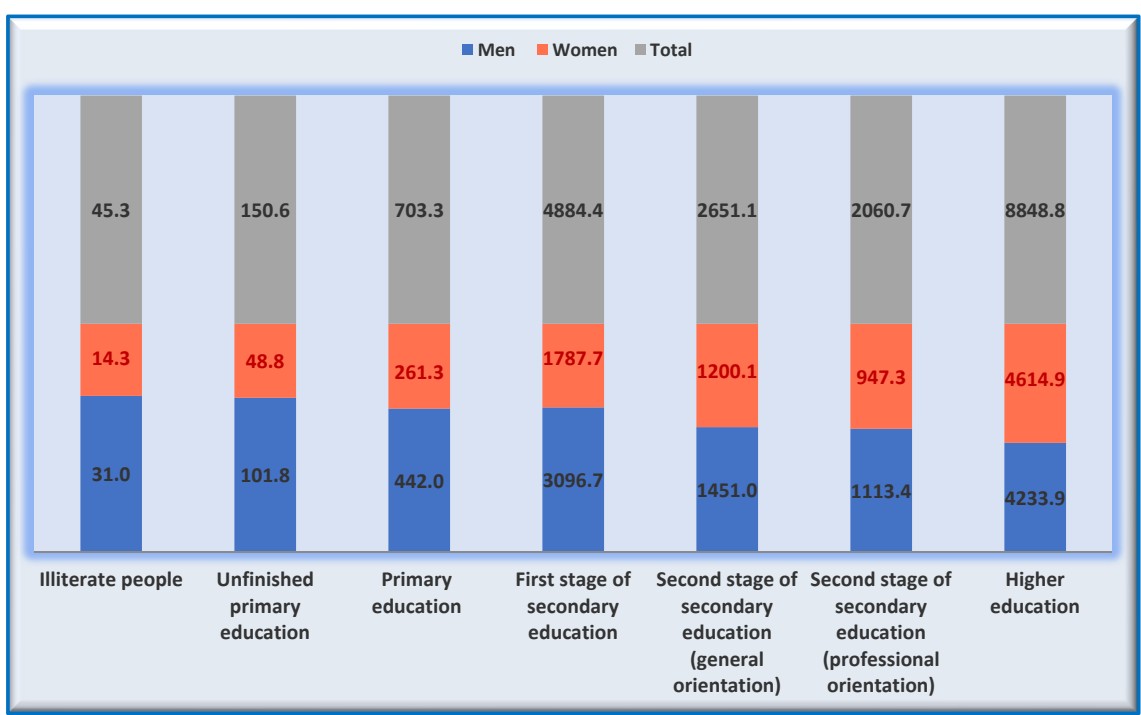

**Figure 7.** Employed persons by sex and educational level attained (in thousands). Source: Authors with data corresponding to the fourth quarter of 2020 provided by the Statistics National Institute (*Instituto Nacional de Estadística*).

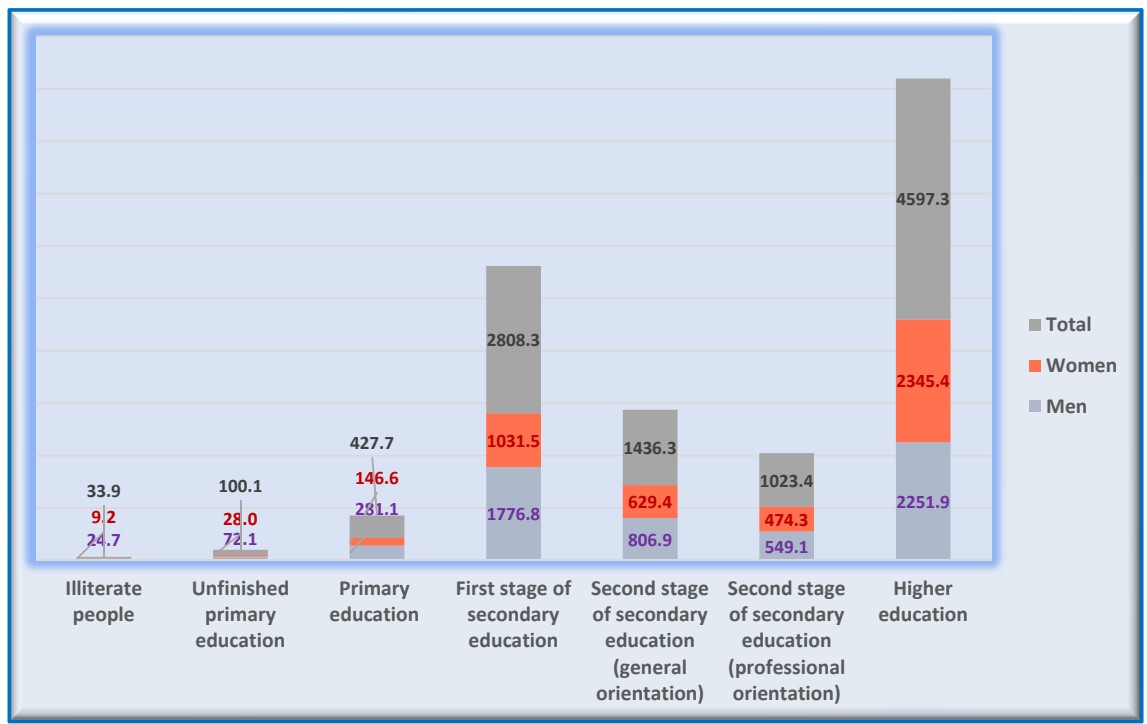

**Figure 8.** Employed married persons by sex and educational level attained (in thousands). Source: Authors with data corresponding to the fourth quarter of 2020 provided by the Statistics National Institute (*Instituto Nacional de Estadística*).

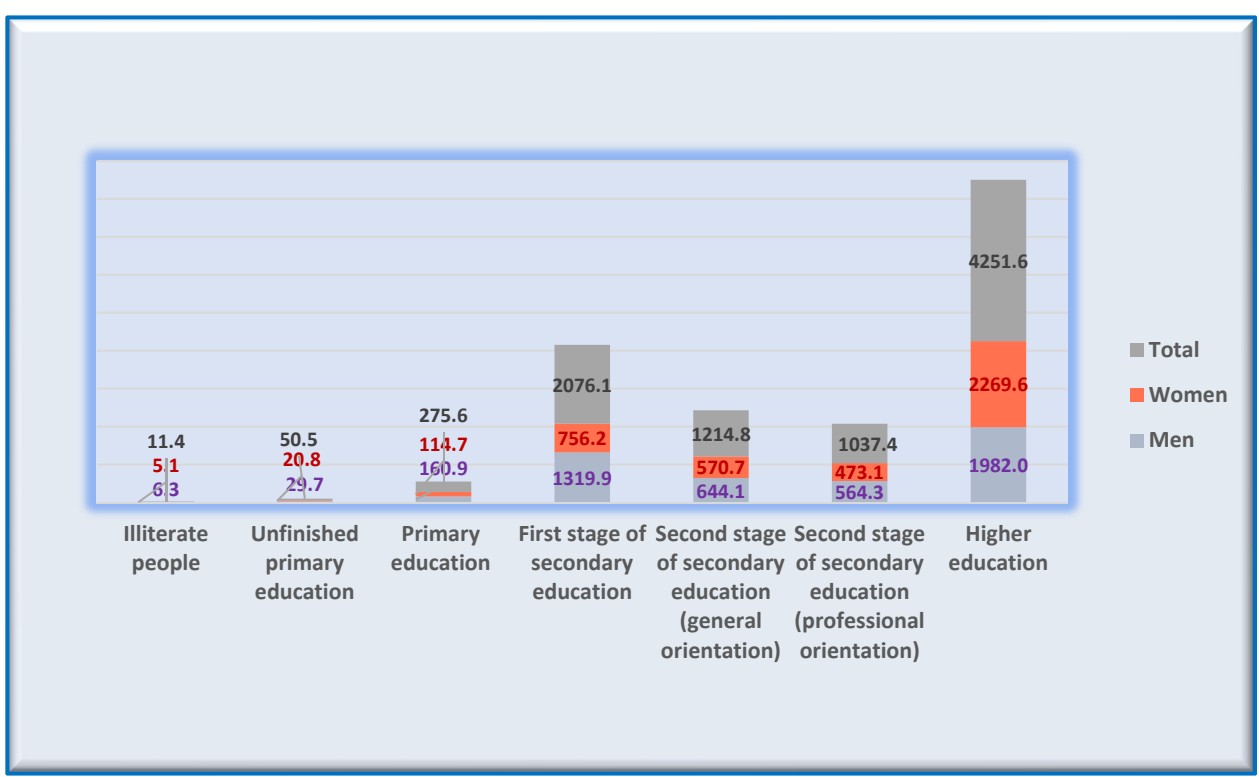

**Figure 9.** Employed unmarried persons by sex and educational level attained (in thousands). Source: Authors with data corresponding to the fourth quarter of 2020 provided by the Statistics National Institute (*Instituto Nacional de Estadística*).

**Table 8.** Average annual income per worker by sex and economic sectors (in Euros).

|  | Men | Women | Difference between Men and Women | Women's Pay Gap Compared to Men (in %) |
|---|---|---|---|---|
| Industry | 29,601.31 | 23,829.62 | 5771.69 | 19.50 |
| Building | 22,997.41 | 21,388.78 | 1608.63 | 6.99 |
| Services | 26,715.74 | 21,498.28 | 5217.46 | 19.53 |

Source: Authors with data corresponding to 2019 provided by the Statistics National Institute (*Instituto Nacional de Estadística*. Note: To calculate the women's pay gap, we have applied the following formula: (average annual salary for men—average annual salary for women): average annual salary for men × 100.

　　The salary gap between men and women is also evident in Figure 10 and Table 9. Thus, based on the latest available data provided by the Annual Salary Structure Survey, we have prepared this figure and table, in which it can be observed that that gap unfortunately had remained quite high in recent years.

　　Nevertheless, in 2019 there was a certain improvement in this gap. Thus, according to the Annual Salary Structure Survey, the wage salary gap between women and men was 19.5% in that year; in other words, it had fallen 1.9 points with respect to the existing gap a year earlier. The slight improvement produced compared to 2018 was due to the fact that women's wages increased more than those of men in 2019 (3.2% compared to 0.7%). However, the truth is that in 2019 women earned on average 5252 euros per year less than men. Given the fact that we have not worked with data for the years 2020 and 2021 (the COVID-19 period), we cannot be sure at this time if this trend towards the decrease in the wage gap for women has remained the same, has improved or has worsened.

　　In this regard, the current socialist government of Spain has applied measures aimed at preventing the destruction of employment and the drop in income of the population in these times of health crisis (Papell 2020; Ruesga-Benito and Viñas-Apaolaza 2021). Such measures, although they have proven to be effective, have not managed to avoid all the negative

effects on employment of COVID-19 (Araújo-Vila 2020; Gómez and Montero 2020). Women are precisely the most negatively affected by this situation, given that compared to men their employment levels are lower and their unemployment rates higher. Therefore, policies that do not address the different realities of men and women will aggravate pre-existing gaps (Salido-Cortés 2021; Solanas 2020).

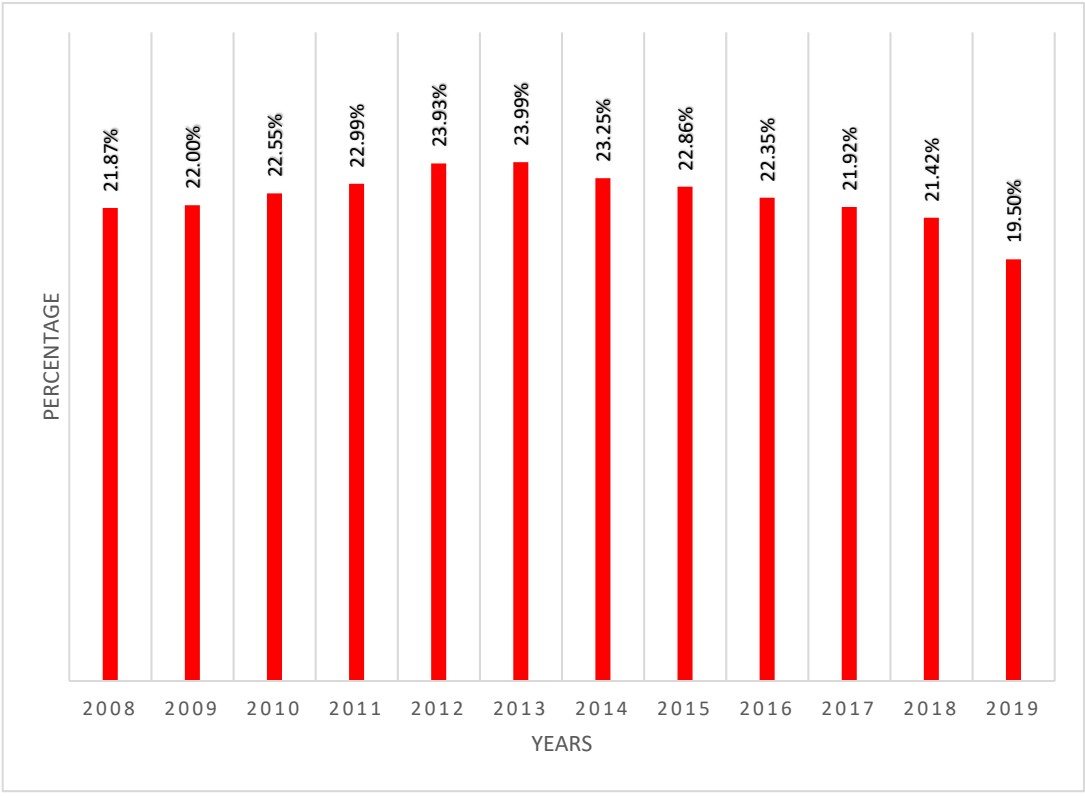

**Figure 10.** Women's pay gap compared to men (period 2008–2019). Source: Authors with data from the Annual Salary Structure Survey provided by the Statistics National Institute (*Instituto Nacional de Estadística*).

**Table 9.** Average yearly wage by sex (in Euros).

| Year | Average Yearly Wage | Men's Average Yearly Wage | Women's Average Yearly Wage | Average Yearly Wage Difference between Men and Women | Men's Average Yearly Wage in Relation to Average Yearly Wage (in %) | Women's Average Yearly Wage in Relation to Average Yearly Wage (in %) | Women's Pay Gap (in %) |
|---|---|---|---|---|---|---|---|
| 2008 | 21,883.42 | 24,203.33 | 18,910.62 | 5292.71 | 110.60 | 86.42 | 21.87 |
| 2009 | 22,511.47 | 25,001.05 | 19,502.02 | 5499.03 | 111.06 | 86.63 | 22.00 |
| 2010 | 22,790.2 | 25,479.74 | 19,735.22 | 5744.52 | 111.80 | 86.60 | 22.55 |
| 2011 | 22,899.35 | 25,667.89 | 19,767.59 | 5900.3 | 112.09 | 86.32 | 22.99 |
| 2012 | 22,726.44 | 25,682.05 | 19,537.33 | 6144.72 | 113.01 | 85.97 | 23.93 |
| 2013 | 22,697.86 | 25,675.17 | 19,514.58 | 6160.59 | 113.12 | 85.98 | 23.99 |
| 2014 | 22,858.17 | 25,727.24 | 19,744.82 | 5982.42 | 112.55 | 86.38 | 23.25 |
| 2015 | 23,106.3 | 25,992.76 | 20,051.58 | 5941.18 | 112.49 | 86.78 | 22.86 |
| 2016 | 23,156.34 | 25,924.43 | 20,131.41 | 5793.02 | 111.95 | 86.94 | 22.35 |
| 2017 | 23,646.5 | 26,391.84 | 20,607.85 | 5783.99 | 111.61 | 87.15 | 21.92 |
| 2018 | 24,009.12 | 26,738.19 | 21,011.89 | 5726.3 | 111.37 | 87.52 | 21.42 |
| 2019 | 24,395.98 | 26,934.38 | 21,682.02 | 5252.36 | 110.40 | 88.88 | 19.50 |

Source: Authors with data from the Annual Salary Structure Survey provided by the Statistics National Institute (*Instituto Nacional de Estadística*).

The gender pay gap is also evident when we focus our attention on the activity sectors. Thus, as Table 10 shows, in 2019 the average annual salary of women was lower than that of men in all the occupations listed in that table.

**Table 10.** Annual average earnings per worker by sex and activity sector in 2019 (in Euros).

| | Men | Women | Difference between Men and Women | Women's Pay Gap (in %) |
|---|---|---|---|---|
| Directors and managers | 60,780.76 | 48,667.65 | 12,113.11 | 19.93 |
| Scientific and intellectual technicians and professionals in health and education | 35,488.04 | 32,136.34 | 3351.70 | 9.44 |
| Other scientific and intellectual technicians and professionals | 40,333.67 | 33,916.08 | 6417.59 | 15.91 |
| Technicians; support professionals | 31,816.30 | 25,950.74 | 5865.56 | 18.44 |
| Office workers who do not deal with the public | 25,634.12 | 20,584.99 | 5049.13 | 19.70 |
| Office workers dealing with the public | 22,437.50 | 18,810.20 | 3627.30 | 16.17 |
| Workers in restaurant and trade services | 18,154.80 | 14,981.95 | 3172.85 | 17.48 |
| Health and personal care workers | 21,005.55 | 15,418.17 | 5587.38 | 26.60 |
| Workers in protection and security services | 29,339.08 | 26,187.39 | 3151.69 | 10.74 |
| Skilled workers in the agricultural, livestock, forestry and fisheries sector | 22,112.47 * | 16,531.69 * | 5580.78 * | 25.24 * |
| Skilled building workers, except machine operators | 21,070.11 | 17,392.78 * | 3677.33 | 17.45 * |
| Skilled workers in manufacturing industries, except plant and machine operators | 24,405.77 | 17,573.21 | 6832.56 | 28.00 |
| Fixed plant and machinery operators, and assemblers | 27,672.25 | 20,327.07 | 7345.18 | 26.54 |
| Drivers and operators of mobile machinery | 20,919.98 | 17,257.64 | 3662.34 | 17.51 |
| Unskilled workers in services (except transport) | 17,346.84 | 12,410.62 | 4936.22 | 28.46 |
| Laborers in agriculture, fishing, construction, manufacturing industries and transportation | 18,920.61 | 16,008.78 | 2911.83 | 15.39 |

Source: Authors with data provided by the Statistics National Institute (*Instituto Nacional de Estadística*. (*) This indicates that the number of sample observations is between 100 and 500, therefore the figure is subject to great variability.

Particularly with regard to the wage distribution, according to the Annual Wage Structure Survey Year (2019), 25.7% of women had in 2019 wages lower than or equal to the Minimum Wage (*Salario Mínimo Interprofesional*), whose annual quantity amounted to

12,600 euros in that year. However, the proportion of men with the same payment level was then 11.1%. This situation of undoubted wage inequality for women was influenced by the highest percentage of women working part-time. If we focus our attention on those who received the highest remunerations, then the pay gap to the detriment of female population was still even more evident in 2019. Thus, while in that year 4.1% of men received wages five times higher than the Minimum Wage, in the case of women only 2.1% of them reached these income levels. However, gender pay inequality is also evident among low-earning workers; that is, those employees whose hourly payment is below 2/3 of the average income. Thus, the proportion of workers with that level of income in 2019 was 15.0%, of which 63.9% were women. One explanation for this lies in the fact that these low salary levels are usually more widespread among part-time workers, of which the majority tend to be women (Annual Wage Structure Survey Year 2019).

In short, the information and data analyzed above show that, at present, there are still clear gender inequalities in Spain. However, the evolution towards greater equality between the sexes that has been taking place in recent decades is encouraging. Undoubtedly, this positive evolution became possible with the arrival of democracy, after the death of dictator Franco, created appropriate legal conditions for progress in this direction. In any case, this progress reflects the crushing slowness of the desirable changes and how much still needs to be done.

One of the challenges still pending in order to achieve gender parity is the creation of truly equal conditions in which men and women can be held jointly responsible for the tasks inherent in caring for their home and children, and that they can do that in a harmonized way with their work. All this applied if we really aspire for the adequate compatibility of work and family life to be shared by men and women on equal terms and not remain an almost exclusively female obligation as has happened, and will continue to happen too frequently.

In any case, the inequalities suffered by women in the workplace are not only due to the fact that they often earn lower wages than men. Thus, a part of these inequalities is due to the fact that there is still an excessive proportion of responsibility positions held by men in the workplace. Such positions, which men have more opportunities to access, involve great decision-making and organizational skills. This is mainly due to the fact that a social mentality or perception is still widespread according to which the duties of caring for children and the home are intrinsically feminine. Consequently, men are or may feel 'liberated' from these obligations and have all their time to devote to their professional careers. These circumstances or social customs mean that a high proportion of women cannot de facto choose (or, even, do not even consider opting) to occupy positions of high responsibility in which the top wages are charged. This is how what is called the glass ceiling is produced; that is to say, a de facto situation that makes it impossible for women to access certain job positions with a high socioeconomic level and an advanced job qualification requirement.

Weighty changes are still needed in order to achieve parity between men and women, not only at the level of social and labor rights (which are already egalitarian in our country) but also and above all at the level of the facts, that is, of the economic, educational and cultural circumstances that continue to maintain the gender gap and the reproduction of the social mentalities that sustain it. Particularly, an ingrained persistence of such mentalities was evident in the last Survey on Quality of Life at Work carried out in Spain in 2010 by the Ministry of Employment and Social Security. From the data of this Survey, we have made Figure 11, which shows the opinion that the interviewees of different sexes had about the possibility of requesting or not a work leave to take care of their family. We can see in this figure how that opinion was different depending on whether the interviewees worked in the private sector or in the public sector.

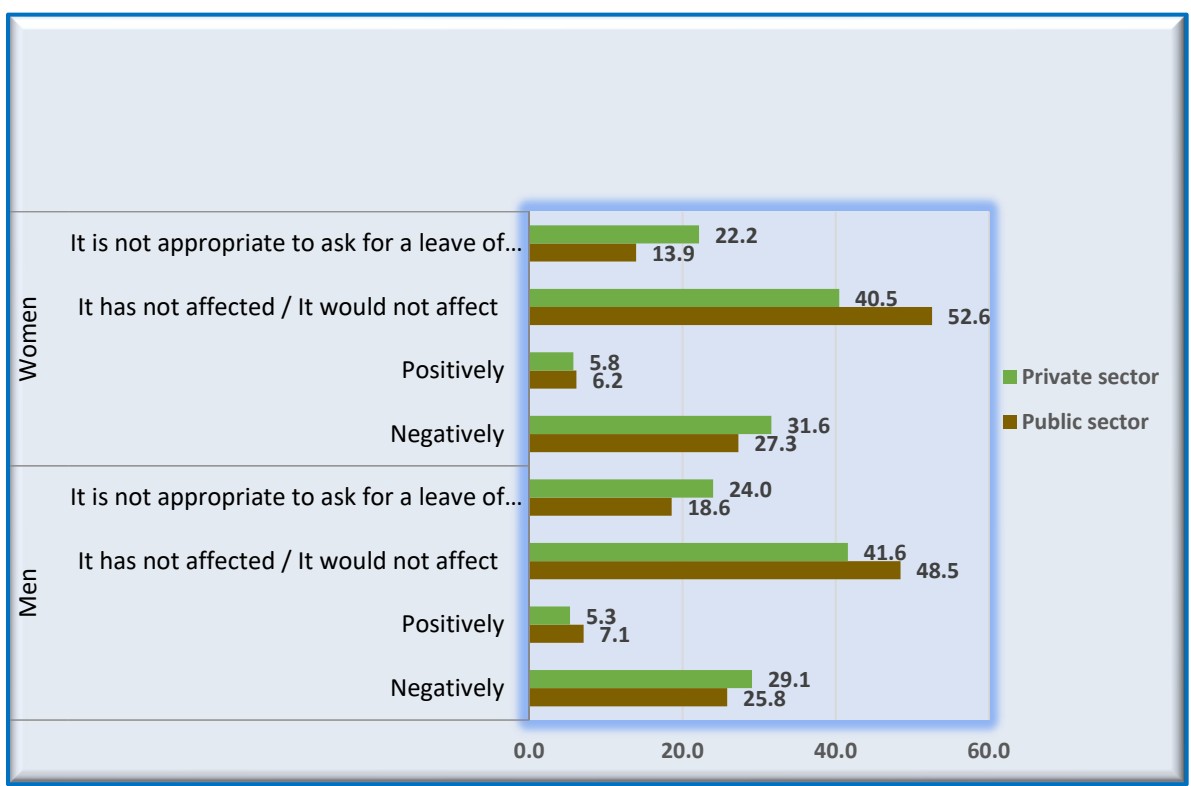

**Figure 11.** How would it affect requesting a level of absence for family reasons according to sex and activity sector (percentage distribution of responses). Source: Authors with data from the 2010 Quality of Life at Work Survey (*Encuesta de calidad de vida en el trabajo*).

Thus, Figure 11 shows that more than half of the women interviewed (52.6% of them) who worked in the public sector thought that taking a leave of absence for family reasons would not affect their professional careers. In contrast, the proportion of women who worked in the private sector with the same opinion was reduced to 40.5% of those interviewed. Additionally, in the case of men, the majority of those interviewed who thought that it would not affect them to take a leave of absence for family reasons (48.5% of them) worked in the public sector, but among private sector employees the proportion of men who thought the same amounted to 41.6%. As can be seen, there are hardly any differences (1.1%) between the percentages of women and men working in the private sector who thought that taking a leave of absence would not affect their professional careers. Nor is the percentage difference (4.1%) between women and men in the public sector who expressed the same opinion very high. This fact, together with the fact that these percentages were clearly below their equivalents in the public sector, could be understood in the sense that both sexes shared a kind of 'business culture'. According to this 'culture', it seems that asking for leave is not looked favorably upon by a noticeable quantity of people who tend to consider something like that exercising the right to take leave of absence entails losing work time.

In relation to the aforementioned, another noteworthy fact from Figure 11 is that, both among public sector workers and those in the private sector, there were more women than men who perceived that taking a leave of absence to care for their family would negatively affect their professional career. Likewise, among both private and public sector workers, the proportion of men who thought it was inappropriate to take a leave of absence for caring for their family was higher than that of women who thought the same. In other words, a large part of these women, in some way, felt that their family responsibility impelled them to ask for that leave of absence. Finally, for both sexes, it is worth highlighting the fact that the proportion of those who considered it inappropriate to take a leave of absence for family reasons was higher among private sector workers than among those who worked in the

public sector. Undoubtedly, apart from the above-mentioned 'business culture' reasons, this is closely related to the fact that, in the public sector, working conditions are generally more stable and less precarious than in the private sector, which facilitates a bigger perception of social protection and greater capacity to exercise labor rights among public sector workers than among those in the private sector.

As has been asserted before, the previous comments have been made based on data that were collected in 2010 through the last Survey on Quality of Life at Work carried out in that year. The sad thing is that, unfortunately, the circumstances have hardly improved since then, as far as overcoming female inequalities is concerned. Thus, as can be seen in Figure 12, made with data from 2020, female employment rates, which are in all cases contemplated in this figure below male employment rates, are higher and are closer to the employment rates of men among women without children. In addition, employment rates are greatly reduced among women with children who due to their age have to be cared for (that is, under 12 years old), but this does not occur among men who have children of the same age. Furthermore, as the number of children of caregiving age increases, the proportion of women working decreases, while in the case of men, this proportion does not undergo significant changes.

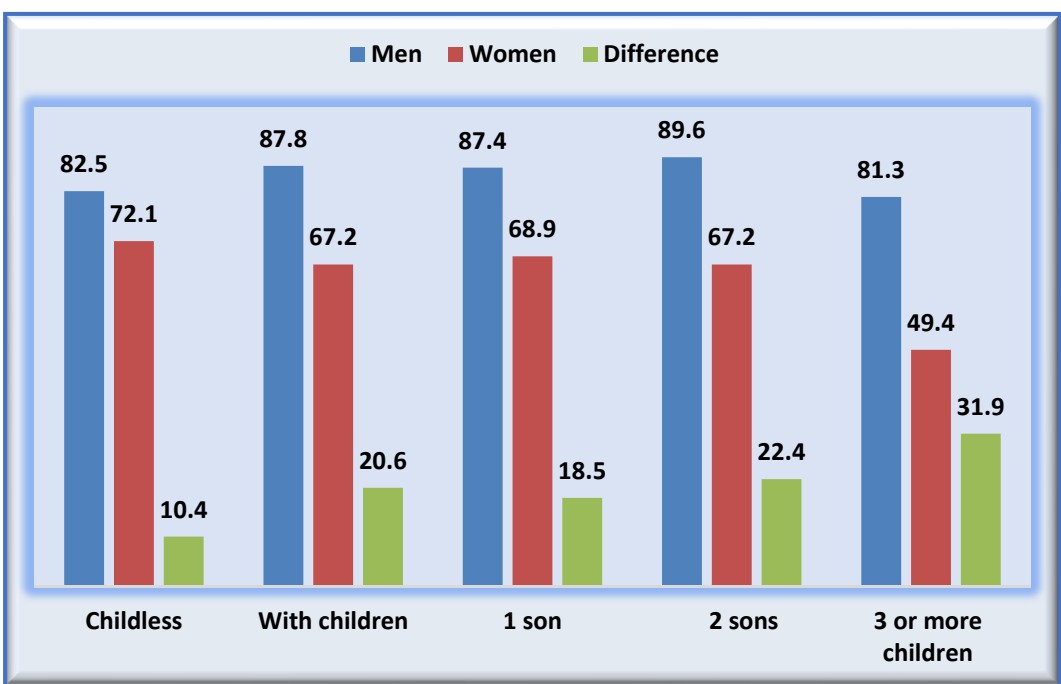

**Figure 12.** Percentage of employment rates by people sex aged 25 to 49 depending on whether or not they have children and the number of children under 12 years old. Source: Authors with data from the Labor Force Survey (*Encuesta de Población Activa*) corresponding to 2020 provided by the Statistics National Institute (*Instituto Nacional de Estadística*).

Unfortunately, the situation reflected in Figure 12 is not something that has only occurred in 2020 but is a highly rooted social phenomenon that makes many women dedicate themselves to caring for their children, and therefore they do not seek work outside from home or even leave that job if they have it. In this way, as can be seen in Table 11, in the period 2011–2019 the lower percentages of employed women with children under 12 years old have been a constant in relation to men with children of the same age. What is more, as the number of children has increased, the percentages of female employment have become lower.

**Table 11.** Employment rates in Spain of adults aged 20–49 years by sex and number of children under 12 years old (in percentages).

| Number of Children Less Than 6 Years | Year | 2011 | 2012 | 2013 | 2014 | 2015 | 2016 | 2017 | 2018 | 2019 |
|---|---|---|---|---|---|---|---|---|---|---|
| 1 | Women | 60.9 | 62.9 | 60.6 | 65.4 | 65.1 | 69.6 | 67.7 | 69.9 | 69.4 |
| | Men | 79.8 | 76.6 | 75.4 | 78.5 | 83.3 | 86.3 | 87.6 | 88.3 | 90.2 |
| 2 | Women | 58.1 | 57.1 | 54.9 | 58.7 | 61.7 | 59.7 | 65.2 | 65.3 | 65.3 |
| | Men | 79.2 | 80.3 | 76.4 | 80.6 | 82.1 | 84.6 | 86.9 | 89.4 | 89.8 |
| 3 or more | Women | 44.5 | 42.6 | 44.8 | 43 | 43.5 | 41.5 | 48.8 | 45.3 | 45.2 |
| | Men | 71 | 67.4 | 61.4 | 69.6 | 72.2 | 71.3 | 77.1 | 81.6 | 79.9 |
| **Number of Children from 6 to 11 Years** | **Year** | **2011** | **2012** | **2013** | **2014** | **2015** | **2016** | **2017** | **2018** | **2019** |
| 1 | Women | 63.7 | 61.1 | 59.3 | 59.4 | 65 | 65.5 | 69.3 | 69.1 | 71.4 |
| | Men | 72.4 | 72.2 | 74 | 76.7 | 78.7 | 81.7 | 84 | 87.8 | 83.9 |
| 2 | Women | 60.3 | 57.1 | 58.3 | 60.4 | 63.9 | 65.1 | 67.3 | 70.0 | 72.4 |
| | Men | 81.3 | 79.7 | 79.1 | 83.9 | 83.2 | 86.4 | 87.5 | 89.9 | 89.4 |
| 3 or more | Women | 52.8 | 53.9 | 53,6 | 52.7 | 61.7 | 51.3 | 56.6 | 59.4 | 59.8 |
| | Men | 73.2 | 67.5 | 74.1 | 76.3 | 77 | 76.8 | 82.4 | 80.3 | 76.5 |

Source: Authors with data from EUROSTAT (available at: https://appsso,eurostat,ec,europa,eu/nui/show,do?dataset=lfst_hheredch&lang=en) (accessed on 23 September 2021).

Thus, recent data available show that women continue to be primarily responsible for caregiving. In this regard, according to 2018 Labor Force Survey (EPA) data, 86.9% of men interrupted their work for a maximum period of six months to devote themselves to taking care of their family (Mujeres en Cifras 2020). Nonetheless, in the case of women, the interruption periods for the same reason were more distributed and, in addition, for a significant part of them, they were longer. So, 49.9% of women interrupted their work for six months, 20.9% between six months and one year and 9.4% between one year and two. In addition, it should be noted that the percentage of women who interrupted their work for more than two years was 17.7%, compared to 2.8% of men.

To a large extent, this is due to the fact that there are still many women who have internalized, as their own and almost exclusive to their gender, the responsibilities of home caregivers. The internalization of this responsibilities by many women, the emotional pressure, and the health problems that this often entails for them are facts that the second author of this article has been able to verify in the interviews carried out on the occasion of his involvement in two previous qualitative studies (Del Río-Lozano et al. 2013; Entrena-Durán et al. 2021).

## 4. Discussion

Both the interviews carried out in the aforementioned previous qualitative researches and the participant observation conducted in the course of them have allowed us to verify that a macho culture or mentality persists in Spain (Borrell et al. 2010; Daros 2014; Mirandé 2018; Pérez-Martínez et al. 2021). This means that a series of roles continue to be socially feminized, despite the fact that these do not have to be unique to women (Borrell et al. 2010; Fernández-Cordón and Tobío-Soler 2019; Glass and Fujimoto 1994; Burin and Meler 1998). Additionally, this macho feminization of certain functions is evident in the mentality of many men. Even the fact of having reached a high educational level does not prevent this mentality from continuing, in such a way that we have verified that, even among young university students, there is a significant sector of them who think or assume that women are responsible for buying and cooking food, which they see as inherently feminine tasks (Entrena-Durán et al. 2021; Fernández-Cordón and Tobío-Soler 2019). This

shows that there are still traces of that patriarchal mentality according to which the major role of women is to care for their family and home (Cerreti and Navarro-Guzmán 2018; Muller-Flury 2021). Regardless of whether this mentality is conscious or unconscious (in most cases it seems to be the latter), the fact is that the persistence of this situation proves that it is still too early to say that patriarchy has ended (Castells 2006; Tobío-Soler et al. 2021). However, the most worrying thing is that there are still a large number of women who express perceptions and behavior habits through which it is noticed that they have ended up internalizing the aforementioned feminization of roles that are not exclusively feminine at all.

This internalization, which entails taking charge of a series of duties and roles that many women end up assuming as inherent in their 'feminine nature', can be understood by resorting to the Bourdesian concept of *habitus*. Thus, this concept is very adequate to explain the women's predisposition to adopt, as their own and inherent to their feminine condition, the role of carrying out the tasks of caring for her family. In this way, women internalize a gender role that is nothing more than a product of history; that is, a social construction that can be deconstructed or modified when the sociocultural situation that makes its production and reproduction possible changes. However, women assume this role as inherent to their gender identity, as if it were innate to their 'feminine nature' and accept that this is in line with their supposed natural psychological aptitudes for such tasks. By acting in this way, women contribute to the production and reproduction of 'male domination' over them. In addition, as a consequence of this, this domination is legitimized as 'normal', since it is assumed something like that it is based on what is presupposed to be the female biological condition (Bourdieu 2001).

According to this same logic, the biological nature of men would equip them better than women to function in the socio-labor life that takes place outside the home. Therefore, a fact whose causes are merely social tends to be seen and legitimized as 'natural', but the truth is that this fact is simply due to the circumstance that men tend to have greater access and control over the material and symbolic resources that sustain their dominance in the public sphere. In these circumstances, many men, and society in general, tend to think something like that, when they have to occasionally take responsibility for certain domestic tasks, they are performing exceptional tasks that are not inherent in their biological masculine nature. As a consequence, men can even improve their self-esteem when they take on these roles; above all, because they tend to achieve social recognition that is based on undoubtedly macho thinking, regardless of the explicit intentions and the gender of those who think so (Mirandé 2018; Pérez-Martínez et al. 2021). This macho thinking could be formulated something like 'what a good person is that man to help his wife with the housework or take responsibility for those tasks when she is not at home or is ill . . . '. These words were spoken by one of the women interviewed in recent research made by the second author of the current article (Entrena-Durán et al. 2021).

Unlike men, women, even when working outside the home (thus entering the public sphere), have fewer opportunities to avoid their role as caregivers in the home as this role has been strongly internalized by the majority of society and of women as an obligation inherent to their feminine gender. As a consequence, they tend to be very self-demanding and to blame themselves when they feel that they have failed in their primary responsibility in cases where they deem that their home is not running well (Del Río-Lozano et al. 2013).

According to the 'Global Gender Gap Report 2021' of the World Economic Forum (Global Gender Gap Report 2021), Spain has ceased to be in the eighth position, in terms of the degree of gender equality achieved, and has fallen to the fourteenth position of the 156 countries included in the ranking of that Report. If we restrict the comparison to the scope of Western Europe and North America, then Spain's position is ninth.

The four dimensions analyzed in the aforementioned Report are: Economic Participation and Opportunity, Educational Attainment, Health and Survival, and Political Empowerment. Of these dimensions, the one that has experienced the greatest decrease in positions, although not in global score, is Health and Survival, in such a way that healthy

life expectancy has been especially affected in the context of the COVID-19 crisis. This is not only due to the direct effects on health of this crisis but also to a series of indirect consequences of it, such as the rise in personal instability and precarious employment, macho violence or the burdens of caring, which, when assumed mainly and almost exclusively by women, have negative effects on their physical and mental health.

The aforementioned is confirmed when one consults study 3312 (Survey on the mental health of Spaniards during the COVID-19 pandemic), conducted by the Spanish Center for Sociological Research (*Centro de Estudios Sociológicos*) on 4 March 2021 (CIS 2021). Thus, this study indicates that, during the pandemic, 17% of women have felt bad for having little interest or pleasure in doing things; 13.5% of them have been depressed or hopeless; 13.2% have felt nervous, anxious or very upset (with the nerves on edge); and 7.9% have felt unable to stop or control their worries. However, the aforementioned figures were reduced, respectively, to 7.6%, 5.2%, 5.5% and 4.2%, when the respondents were men.

In particular, as the Global Gender Gap Report shows, it is in the economic sphere where parity between women and men has been most threatened as a result of the COVID-19 crisis. In this way, the gap in estimated earned income has widened in the last year. To a great extent, this has been due to the increase in part-time employees and unemployed women compared to the previous year's Report. In the particular case of Spain, it is worrying that this inequality of women in the economic sphere could increase in the future since many of tomorrow's jobs will be in the digital sphere, in which the proportion of men continues to be strongly predominant (Fernández-Montes 2021). Thus, in our country, according to 2020 data provided by the Spanish National Institute of Statistics, men have a presence in the digital sector of 70.66%, while that of women is 29.33%.

The field of education continues to be Spain's strong point according to the Global Gender Gap Report since it is in this field where we are closest to gender parity. The detail that the literacy rate score has dropped is significant. Thus, this rate has gone from 0.998 in 2020 to 0.990 in 2021. Perhaps this decrease is related to the fact that the COVID-19 health crisis has led to the death of more older men than older women, who for a long time did not have the same opportunities as men to access education. Despite the continuous and evident advances with respect to the equal access to the different educational levels among the younger Spanish generations, it is needed to point out that the success rate among women in the STEM careers (Science, Technology, Engineering and Mathematics, for its acronym in English) is 12.44%; that is, a percentage clearly lower than that of men, which is 37.34%.

In any case, it is encouraging that, according to Eurostat data, Spain is among the countries of the European Union of 27 (EU-27) with most women scientists. Specifically, it is the 3rd country in the EU-27 in this regard and the 5th with the highest parity between men and women dedicated to science. Thus, of the 1,506,200 people that make up the scientific community in Spain, 743,100 are women. This amounts to 49.3%, which is above the European average of 41%. Only Lithuania (55%), Latvia (52.7%), Denmark (51.7%) and Bulgaria (50.1%) surpass Spain in terms of the balance between the number of men and women working in science and technology. On the opposite side are Luxembourg (28%), Finland (31%) and Hungary (32.6%), countries in which the number of male scientists still far exceeds that of women. Regarding the total number of women devoted to science, Spain is surpassed only by Germany, with 1,120,000 female scientists, and France, with 810,800. However, there is still a way to improve in our country, where the female presence in technical careers is still 30% (Spain Among EU Countries with Most Women Scientists 2021).

Last but not least, one must bear in mind the worrying fact that Spain's score in the Political Empowerment dimension has fallen from 0.527 in 2020 to 0.491 in 2021. This fall in turn has led to a 7-point drop for Spain (it has gone from position 8 to 15) in the ranking of countries included in the Global Gender Gap Report for 2021. In part, this worsening of the situation is due to a decrease in the percentage of women deputies in parliament. Thus, while in the Global Gender Gap Report 2020 women held 47.4% of the seats, currently that percentage has dropped to 44.0% (Global Gender Gap Report 2020).

## 5. Conclusions

A characteristic feature of this article is that in it we have analyzed the issue of gender parity in Spain from various perspectives. Thus, we have combined the study of legal regulations aimed at promoting gender equality with statistical analyzes about the real salary and employment gaps of women. All this, together with the reanalysis that we have made of the qualitative research mentioned above, has allowed us to achieve a global and all-encompassing approach to the object of study. In this way, we have been able to see how, despite the legal advances that have happened and continue to take place in gender parity, worrying inequalities regarding women still need to be overcome with regard to their incorporation into the labor market and salary levels. Likewise, the aforementioned qualitative research has made it possible for us to consider some of the cultural obstacles that still persist and contribute to slowing down the necessary transition towards achieving gender parity. In particular, we have talked about the traces of a macho mentality and sexism (Borrell et al. 2010; Daros 2014; Mirandé 2018; Pérez-Martínez et al. 2021), and what Bourdieu called 'male domination' (Bourdieu 2001). For instance, according to this 'domination', there would still be a significant number of women who would have 'naturalized' and assumed their situation of inequality mentally as 'normal'. Thus, they would have internalized and accepted as inherent to their gender a series of care roles, which should not be their sole responsibility but shared equally by men and women (Entrena-Durán et al. 2021).

A hopeful fact is that the data and analysis that have been offered so far indicate that, despite the fact that there is still a large gap between men and women in a setting as significant as the labor market, the trend towards a gradual decrease in gender inequality seems clear. However, much remains to be done and the progress achieved, far from being consolidated, may be threatened by any unforeseen eventuality. In fact, the crisis triggered by the COVID-19 pandemic has exacerbated the existing inequalities between women and men in aspects such as gender violence, employment, and informal and/or unpaid care. (Hupkau and Victoria 2020). As a consequence, there has been regression or at least stagnation regarding different types of progress that have been achieved in recent decades.

From what has been stated in the preceding pages, it can be deduced that, despite the progress made and the laws and plans implemented in favor of gender equality, there is still much to be achieved. In this sense, as Octavio Salazar has pointed out (Salazar 2012), it is necessary for institutions and policies to continue making efforts to create the right conditions so that the legislative provisions on gender equality that have already been enacted, and those that in the future may enacted, are put into practice with the maximum possible efficiency and results.

Apart from continuing to work to improve educational, political, employment and economic opportunities for women, one of the challenges that remains is to eradicate the more or less macho mentalities that have been entrenched for so many years. This implies working so that all those social perceptions and habits based on gender egalitarian views become increasingly established in the social and labor spheres (Lamolla 2020; Tobío-Soler et al. 2021). This entails encouraging deep and continuous changes in those cultural forms and habits that legitimize and support gender inequalities. Such inequalities are manifested and materialized, above all, in the ways in which the assignment and attribution of roles and responsibilities to men and women take place.

Identifying and analyzing the social perceptions behind such roles' assignment and attribution is an inescapable intellectual task in which we plan to get involved in our next piece of research. In this study, we will do qualitative interviews with men and women aimed at deepening the understanding of their social perceptions about the social distribution of roles by sex. We will do this because it seems obvious that this type of inquiry will help us to be in a better position to understand the way in which the production and reproduction of unequal gender relations and attitudes take place. Such modes of production and reproduction are based on interests and viewpoints of reality that, even though there is an increasingly widespread feminist mentality, are very difficult to change. To a very large extent, these difficulties are due to the fact that such change requires a

great effort that is not always carried out and, when it is made, it is often not stronger than the greater or lesser resistance to change that exists. In fact, such resistance is usually manifested among a large part of people, in such a way that it could be affirmed that even the authors of this work might not be safe from it.

In relation to the aforementioned resistance to change, authors such as Rodríguez, Peña and Torío (Rodríguez et al. 2010) pointed out that, since women have been relegated to the home for so long, even they are reluctant to abandon their domain of it, while many of them have ended up internalizing the idea that men's dominance necessarily relates to the public sphere.

As long as these circumstances last, it can be asserted that working women have a double workload since they have to combine their unpaid homework with work inherent to their job outside the home (Durán-Heras 2013; Nava-Bolaños 2014). All of this is because, as Raquel Lucas has pointed out (Lucas 2012), the role of many men in the home continues to be only that of mere collaborators. This function does not usually harm the life projects of these men since among their roles the care of the home and children is not found as a central responsibility.

**Author Contributions:** I.Á.-P. and F.E.-D. carried out all the stages of the paper: conceptualization, conception, design, research, analysis and conclusions, writing, and final review. All authors have read and agreed to the published version of the manuscript.

**Funding:** This research, which is part of a PhD thesis written by Isabel Árbol-Pérez and supervised by Francisco Entrena-Durán, has been carried out at the University of Granada (Spain) with a 4-year predoctoral contract for the training of the University Teaching Staff (*Contrato predoctoral para la Formación del Profesorado Universitario (FPU)*).

**Institutional Review Board Statement:** Not applicable.

**Informed Consent Statement:** Not applicable.

**Data Availability Statement:** The data presented in this study are available on request from the authors.

**Acknowledgments:** The predoctoral contract mentioned in the previous paragraph was founded by the Spanish Ministry of Research and Universities.

**Conflicts of Interest:** The authors declare no conflict of interest.

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
