# Peer review of "Gender Parity in Spain: Attainments and Remaining Challenges"

_socsci, doi:10.3390/socsci11010004_

Round 1

Reviewer 1 Report

Summary:

In this paper authors attempt to analyses the progress in terms of gender parity made in Spain. Firstly, progress in the legislative framework is presented, then statistical data are presented, including differences in activity rates, employment rates and earnings of men and women in Spain, finally in the Discussion authors present female and male perception about taking a leave based on the previously conducted qualitative research and differences in employment rates of men and women without and with 1, 2, 3 or more children under 12 years of age.

The main results are that despite the undoubted legislative advances achieved, the available data show persistence of clear gender inequalities in Spain.

I find the topic of the paper really interesting and I believe that every discussion on the topic of gender parity or gender equality is valuable. However, I think this paper needs to be rethought and change what the paper claims to have shown. This papers’ aims are worthy and I would dearly like to see it succeed in meeting them, but I cannot be sure that is possible.

I have given some comments and suggestions that I believe will improve the quality of the paper but a lot of things needs to be improved to make the paper publishable as a scientific one. In the following, I have divided my comments and suggestions into major issues and more minor stuff (where some of the former are relatively easily fixable—but I still consider them major issues in the below, because of their importance).

Major issues

  1. In Section 2 Materials and Methods it is written:

“…From these data we draw up a series of figures and tables that 130 allow us to show that, despite the undoubted legislative advances achieved, the available 131 data corroborate the persistence of clear gender inequalities in Spain…”,

  • I wouldn't so strongly claim this before showing the data, tables and figures. I would leave all the claims for the section Results or Discussion. So, I suggest reformulating the sentence somehow, for example: "From these data we will draw up a series of figures and tables that will allow us to analyse if undoubted legislative advances achieved are followed by gender equality in Spain or not."

  1. Please give an explanation why you have chosen to evaluate progress in gender equality by using data on activity rates, employment rates and gender pay gap in Spain.

  1. In Section 3.2. a lot of statistical data are presented using tables and figures but without any “deeper” explanations that would give a real scientific contribution to the paper. For example:
    • In Section 3.2., page 7, line 306 – 309 it is written:

“…This is closely related to the fact that there are still a large number of women 306 who are neither working nor looking for work outside the home, since they are devoted 307 to the social role that has traditionally been assigned to the female gender as housewives 308 and/or mothers…”

  • Where is the evidence (any reference) for this? I suggest looking at Eurostat data on reasons for inactivity: http://appsso.eurostat.ec.europa.eu/nui/show.do?dataset=lfsa_igar&lang=en

  • In Section 3.2., page 8, line 322 – 325 it is written:

“…However, it is 322 possible to observe a more pronounced decrease in the difference between the activity 323 rates of men and those of married women, in such a way that, while in 2009 this difference 324 was 16.37 points, in 2020 it has been reduced to 8.46 points…”

  • How you explain this when it is expected that married women would be even more overburdened with unpaid work?

  • In Section 3.2., page 8, line 343-347 is written:

“…Could this higher proportion of separated or divorced women who work or look for job be understood in the sense that these women are more pushed to look for an employment because they do not have the financial support of a partner? Or, could this situation 345 reveal that there are a considerable number of women who, precisely because they 346 choose to enter the labor market, renounce (or, rather, they are compelled to renounce 347 due to circumstances) from marriage?...”

  • I would add this: or only financially independent women are ready to get out of the unhappy marriage while financially dependent women choose to stay in unhappy marriage because they don't have an alternative?

  • In Section 3.2., page 16, line, lines 547 - 548

“…However, it can be seen that, with regard to unmarried employees (figure 8), the difference between employed men and women in the same work situation is less than in the case of married employees…”

  • How you explain this?

  1. There are a lot of confusing sentences, for example:

On page 9, line 380 – 381: “We make this clarification given that next, in addition to commenting on figure 5, we present a series of additional data and calculations that are not reflected in said figure, at the same time we make several tables on that data.”

On page 22, line 784-786“The aforementioned is confirmed when one consults study 3312 (Survey on the mental health of Spaniards during the COVID-19 pandemic), conducted by the Spanish Center for Sociological Research (Centro de Estudios Sociológicos, CIS, in Spanish) on March 4, 2021[15]…”

  • On page 22, line 817 put “3rd” instead of “3th”
  • I recommend having somebody else, preferably a native English-speaker (I am a non-native English speaker myself, so I try to also follow my own advice here myself!) reading through the entire manuscript before a possible resubmission of this manuscript.

  1. For figure 6, 7 and 8 I suggest using another type of the chart: so called stacked column chart (horizontal or vertical) with the columns showing the total number of employed persons based on the level of education and then with colours you will present the ratio of men and women in the total value.

  1. I suggest calculating gender pay gap as a percentage, to be exact as a ratio of women earnings and men earnings and comment the difference in percentage points. Usually it is done this way. Then you will be able to add to Figure 9 pay gap for years before 2019 - same that you did for employment and activity rates and show the trend of increase of decrease.

  1. In Section 3.2., page 16, line 566 -567 is written:

“…According to the aforementioned Annual Salary Structure Survey of 2019, the yearly salary gap between women and men was 19.5% in that year; in other words, it had fallen 1.9 points with respect to the existing gap a year earlier…”

  • How did you get this salary gap?

  1. In the Section 3.2., page 17, line 572 – 573 is written:

“…As can be seen in table 8, in the cases of three important economic sectors such as industry, building and services, the average annual earnings per worker for women was lower than the average annual earnings for men in 2019…”

  • Why are these sectors important, i.e. why did you single them out?
  • To make a point use gender pay gap in percentage because then it can be seen if the difference in earnings of women and men in these sectors are higher or lower than the average;
  • I suggest adding the differences in earnings of men and women in female branches and male branches. Usually the differences are even more pronounced in female sectors.

  1. In Section 4 authors mention two interviews carried out, but only one question from one interview was analysed and other are just statistical data on employment rates of men and women without, one, two, three or more children under the age of 12. Please be careful with that and give more information about the conducted research to which you refer.
    • This type of data is also available on Eurostat which allow you to analyse data for longer periods: https://appsso.eurostat.ec.europa.eu/nui/show.do?dataset=lfst_hheredch&lang=en

  1. In Section 4, page 20, line 714 – 715 authors state:

“… Both the interviews carried out in the aforementioned previous qualitative studies and our own participant observation have allowed us to verify that the macho culture or mentality still persists in Spain…”

  • In the paper macho mentality is verified by only one question which is not enough, especially because the perception about taking a leave as a father or mother are almost similar.
  • Besides, what mentioned participation observation has been conducted?

  1. In Section 4, page 20, line 719 – 722 authors state:

“…Even the fact of having reached a high educational level does not prevent this mentality from continuing, in such a way that we have verified that, even among young university students, there is a significant sector of them who think or assume that women are responsible for buying and cooking food, which consequently they see something of an inherently feminine task….”

  • How did you verify the aforementioned?
  • The literature you are referring to is from 2001 and 2006?

  1. In Section 4, page 21 authors state:

“…As a consequence, men can even improve their self-esteem when they take on these roles; above all, because they tend to achieve social recognition that is based on undoubtedly macho thinking, regardless of the explicit intentions and the gender of those who think so. This macho thinking could be formulated something like ‘what a good person is that man to help his wife with the housework or take responsibility for those 760 tasks when she is not at home or is ill ...’.”

“…Unlike men, women, even when working outside the home (thus entering the 762 public sphere), have fewer opportunities to avoid their role as caregivers in the 763 home, as this role has been strongly internalized by the majority of society and of 764 women as an obligation inherent to their feminine gender. As a consequence, they 765 tend to be very self-demanding and to blame themselves when they feel that they 766 have failed in their primary responsibility in cases where they deem that their home 767 is not running well…”

  • Reference is missing for these statements!

Minor issues

  1. I suggest to reformulate Keywords how follows: Spain; Gender equality legislation; Gender inequalities; Activity rates; Employment rates, Gender pay gap, Gender roles, Macho mentality

  1. Text in Spanish should be Italic and avoid the term “in Spanish”;

  1. I suggest avoiding a term "said" in the paper. Firstly, the paper is written so maybe it would be more appropriate to write: "mentioned", "aforementioned" etc. Secondly, I believe other terms will be more suitable for the language of scientific paper

  1. In the Section 3.1., page 5, line 218-220, it is written:

“…In this way, the actions of 218 said government have been characterized by a series of Spanish public policies aimed at 219 promoting gender equality…”

  • Name these policies, at least some of them.

  1. In the Section 3.1., page 6, line 273-276, it is written

“…Said Commission would be responsible for coordinating, monitoring and preparing gender impact reports, as well as making periodic evaluations on the effectiveness of the actions carried out and meeting at least twice a year…”

  • What was written in those evaluations regarding the effectiveness of the actions carried out?

  1. In the Section 3.2., page 9, line 383 – 384 is written:

“…First, figure 5 shows that the activity branches in which the majority of jobs are male are: Building, Extractive industries, Transport and storage, Agriculture, cattle raising, forestry and fishing, Manufacturing industry, and Information and communications...”

  • I suggest using Uppercases for all branches or for none

  1. In Section 3.1. evaluate government measures during covid 19 and its effects on gender equality? Any budget cuts that could affected negatively or positively on gender equality and similar? Did government do anything to support women during pandemic?

  1. In section 3.2., page 6, line 295 – change “devoted” into “devote”

  1. In section 3.2., page 6, line 297 – exclude “other”

  1. In Section 3.2., page 7, line 307 – put “ratio” instead of “relationship”

  1. In section 3.2., page 9, line 357 – 360 is written:

“…Apart from what has been said before, we take into account the employment rate, which is the quotient between, on the one hand, the number of people employed included in the age range from 16 to 64 years, and, on the other, the whole population of the same age range; that is, the working-age population….”

  • I suggest putting “ratio” instead of “quotient”

  1. In the sentence on page 9, line 378 -379:

“…The reason for this exclusion has been to prevent this figure from being larger and more detailed than it already is, since that would have made it difficult to see and understand it…”

  • Exclude “to see”
  1. I suggest putting Table 1, 2 and 3 before Figure 5. Do the same with the explanations of those tables, i.e. explain first those tables and then focus on Figure 5 (including everything about it).

  1. In the titles of figures and tables avoid term: “thousands of people” and use “in thousands”

  1. In Figure 8, in legend translate “Mujeres” into “Women”

  1. Always use same colours for men and same for women because by switching colours in different figures you are confusing readers (for example Figure 9)

  1. In Section 3.2., page 17, line 593:

“…Thus, while in that year 4.1% of men received wages five times higher than the SMI, in the case of women only 2.1% of them reached these income levels…”

  • Avoid Spanish acronyms

Author Response

Dear Reviewer 1,

Thank you very much indeed for your careful reading of our manuscript and the detailed recommendations you make to us. Your contributions have been a great challenge for us and have stimulated us a lot in our laborious and exciting task of reforming and rewriting the manuscript. As you can see, we have inserted new tables and figures, while we have improved the appearance of the latter. Likewise, we have added numerous new references related to our object of study and we have written new paragraphs and modified others. In short, we believe that, thanks to having taken all your recommendations into account, the current version of the manuscript has improved significantly compared to the first one.

We are attaching you a pdf file with our answer to your suggestions.

Best wishes,

The authors.

Reviewer 2 Report

Dear authors,

Please read my comments below:

This is an interesting article with a lot of relevant statistical information. The 'quantitative' methodology used was based on descriptive statistics and frequencies, which makes the robustness of the method less. However the content is interesting. I would like to see a discussion with a larger number of scientific papers cited. Did you not put them because they do not exist or because you did not look for them?
There is little background literature in the Introduction which is also the theoretical framework at the same time. I would like to see more studies reviewed.

Best Regards and good luck.

Author Response

Dear Reviewer 2,

We are attaching you a pdf with our answer to your suggestions. Best Wishes.

Reviewer 3 Report

Dear Authors 

Your article is interesting, and I am grateful for the opportunity to read it. The subject of the research is interesting, and the results of the research give a lot of new information and possibilities of further analysis. However, I have a few more significant challenges with the paper. From my point of view, there are some revisions the authors should consider improving the paper. 

  • It would be appropriate to specify in more detail how this research differs from the already published paper that deals with a similar topic. To increase the significance of the results, the discussion part should embrace the differences and similarities among your findings and those of other scholars. 
  • Conclusions should be expanded, pointing to the limitations of the analyzed problem and defining the directions of further research. 
  • The quality of the figures is not sufficient. 

Author Response

Dear Reviewer 3,

We are attaching you a pdf with our answer to your suggestions. Best wishes.

Round 2

Reviewer 1 Report

I think that you have done a very fine job in addressing all my previous comments and suggestions -- and so I am now very happy to support acceptance of this manuscript, on my end.